# Tackling Infectious Diseases in the Caribbean and South America: Epidemiological Insights, Antibiotic Resistance, Associated Infectious Diseases in Immunological Disorders, Global Infection Response, and Experimental Anti-Idiotypic Vaccine Candidates Against Microorganisms of Public Health Importance

**DOI:** 10.3390/microorganisms13020282

**Published:** 2025-01-27

**Authors:** Angel Justiz-Vaillant, Sachin Soodeen, Darren Gopaul, Rodolfo Arozarena-Fundora, Reinand Thompson, Chandrashekhar Unakal, Patrick E. Akpaka

**Affiliations:** 1Department of Para-Clinical Sciences, University of the West Indies, St. Augustine Campus, St. Augustine 330912, Trinidad and Tobago; sachin.soodeen@my.uwi.edu (S.S.); reinand.thompson@sta.uwi.edu (R.T.); chandrashekhar.unakal@uwi.edu (C.U.); eberechipatrick.akpaka@uwi.edu (P.E.A.); 2Department of Surgery, Morehouse School of Medicine, Atlanta, GA 30310, USA; dgopaul@msm.edu; 3Eric Williams Medical Sciences Complex, North Central Regional Health Authority, Champs Fleurs 330912, Trinidad and Tobago; rodolfo.arozarenafundora@uwi.edu; 4Department of Clinical and Surgical Sciences, Faculty of Medical Sciences, The University of the West Indies, St. Augustine 330912, Trinidad and Tobago

**Keywords:** HIV, nosocomial infections, antibiotic resistance, salmonella, blood-transfusion safety, immunology, West Indies, mycobacterium tuberculosis, Gram-negative bacteria, IgM antibodies, anti-idiotypic vaccines

## Abstract

This paper explores various aspects of microbiology and immunology, with a particular focus on the epidemiology and molecular characterisation of infectious diseases in the Caribbean and South America. Key areas of investigation include tuberculosis (TB), experimental vaccines, and bloodborne pathogens. A retrospective study conducted in Jamaica highlights the significance of early HIV screening, timely diagnosis, and inte-grated care. The paper also examines the challenges posed by nosocomial infections, particularly those caused by antibiotic-resistant Gram-negative bacteria and methicillin-resistant *Staphylococcus aureus* (MRSA), emphasising the critical importance of infection control measures. Additionally, it explores the regional microbiome, the global response to infectious diseases, and immune responses in patients with immunodeficiency disorders such as severe combined immunodeficiency (SCID) and chronic granulomatous disease (CGD), underscoring their heightened susceptibility to a wide range of infections.

## 1. Introduction

Nosocomial infections, which are a major concern in healthcare settings, are examined in this study. These infections, caused by drug-resistant pathogens like Gram-negative bacteria and methicillin-resistant *Staphylococcus aureus* (MRSA), are prevalent in Trinidad and Tobago. They lead to increased morbidity, mortality, and healthcare costs, underlining the importance of effective infection control measures and the prudent use of antibiotics [1,2].

Another critical issue addressed in this paper is foodborne illnesses, particularly salmonella infections linked to poultry. This study evaluates the prevalence of salmonella in Jamaica’s poultry industry, examining the implications for food safety and public health. Effective control of salmonella requires stringent biosecurity measures and public education. In addition to bacterial infections, the safety of blood transfusions is of paramount concern.

Cabrera et al. conducted a systematic review on ageing with HIV in Latin America and the Caribbean, exploring its health, social, and economic implications. The review highlighted challenges such as comorbidities, stigma, and limited healthcare access affecting older adults with HIV. The findings emphasise the need for tailored interventions, improved healthcare systems, and regional collaboration to support this population effectively [3].

This multidisciplinary study provides insights into tackling infectious diseases through microbiological and immunological advancements. Infectious diseases continue to pose significant public health challenges worldwide, particularly in resource-limited regions like the Caribbean and Latin America. Microbiology and immunology play a crucial role in understanding the mechanisms of disease transmission, pathogenesis, and host immune responses. The research by Poteat et al. (2018) discusses the broader epidemiology of HIV, sexually transmitted infections (STIs), viral hepatitis, and tuberculosis among incarcerated transgender individuals and touches upon regional disparities, including the Caribbean and Latin America. While the article focuses on the global burden and limited data availability, it references the Caribbean region and Latin America as areas where there are notable gaps in research [4].

In addition, we investigated the microbiome, which encompasses diverse microorganisms and their genetic material, and is crucial to health, disease prevention, and environmental stability. In Latin America and the Caribbean, its study is vital for addressing unique regional challenges, including infectious diseases, malnutrition, and environmental degradation. Understanding microbiome dynamics offers transformative opportunities for improving public health, sustainable agriculture, and environmental resilience across these regions.

This review will explore critical themes in infectious diseases, focusing on nucleic acid testing (NAT) and its applications, infectious diseases in the context of immunological disorders, and global responses to emerging infections. It will also address the unique challenges of tuberculosis and HIV screening in low-resource settings, highlighting the ethical, practical, and diagnostic barriers in such environments. By examining these interconnected topics, this review aims to provide a comprehensive understanding of current advancements, challenges, and strategies in infectious disease management and their implications for global health, particularly in resource-constrained settings where disease burdens are often highest and protective anti-idiotypic vaccines.

## 2. Nosocomial Infections Globally

### 2.1. Nosocomial Infections

A nosocomial infection, also known as a hospital-acquired infection (HAI), is an infection that a patient acquires during their stay in a healthcare facility, such as a hospital, clinic, or long-term care facility. These infections are not present or incubating at the time of admission and typically manifest 48 h or more after hospitalisation. Nosocomial infections can also occur after discharge if the infection was contracted during the hospital stay [5,6,7].

Common CausesNosocomial infections are caused by various pathogens, including the following:Bacteria (e.g., *Escherichia coli*, *Staphylococcus aureus*, and *Pseudomonas aeruginosa*);Viruses (e.g., influenza and norovirus);Fungi (e.g., *Candida* species).Common Sites of InfectionUrinary tract infections (UTIs): often associated with the use of urinary catheters;Surgical site infections (SSIs): occur at the site of surgical incisions;Pneumonia: particularly ventilator-associated pneumonia (VAP);Bloodstream infections (BSIs): often linked to intravenous catheters.Risk FactorsProlonged hospital stays;Use of invasive devices (e.g., catheters and ventilators);Surgery or other invasive procedures;A weakened immune system (e.g., due to chronic illness or medication);Poor infection-control practices.PreventionStrict adherence to infection-control protocols, such as hand hygiene and sterilisation;Appropriate use of antibiotics to prevent resistance;Regular cleaning and disinfection of hospital environments;Minimising the use of invasive devices and removing them promptly when no longer needed.

Nosocomial infections can lead to significant health complications, extended hospital stays, and increased healthcare costs, making their prevention a critical aspect of patient safety.

Wade et al. conducted a mixed-methods systematic review on healthcare-associated infections and antibiotic prescribing in CARICOM hospitals. The study revealed high infection rates, inappropriate antibiotic use, and inadequate infection-control measures. The authors emphasised the urgent need for antimicrobial stewardship programmes and improved infection prevention strategies to address these challenges in the region [5].

Ponce de Leon et al. conducted a systematic review and meta-analysis on *Pseudomonas* infections among hospitalised adults in Latin America. The study identified high prevalence rates, antimicrobial resistance, and significant mortality associated with these infections. The findings underscore the critical need for enhanced infection control measures and antimicrobial stewardship in the region to mitigate the impact [6].

García-Betancur et al. provided an updated review on the epidemiology of carbapenemases in Latin America and the Caribbean. The study highlighted a rising prevalence of carbapenemase-producing organisms, with significant geographical variability and associated high mortality rates. Common enzymes included KPC, NDM, and VIM. Contributing factors included the overuse of antibiotics, limited surveillance, and inadequate infection-control measures. The authors emphasised the urgent need for improved regional surveillance, infection prevention programmes, and stricter antimicrobial stewardship policies to combat the spread of carbapenemase-producing pathogens and reduce their clinical and economic burden in the region [1].

Delva et al. conducted an integrative review of hand hygiene practices in Caribbean and Latin American countries. The study revealed inconsistent compliance with hand hygiene protocols, largely influenced by resource limitations, inadequate training, and cultural factors. It highlighted the critical role of effective education, access to hygiene supplies, and institutional support in improving practices. The authors emphasised the need for region-specific strategies to strengthen hand hygiene adherence, reduce healthcare-associated infections, and promote a culture of safety in clinical settings, ultimately improving patient outcomes across these regions [7].

Nagassar et al. examined antimicrobial resistance (AMR) in the Caribbean, highlighting its growing prevalence and significant public health impact. The study identified widespread resistance to commonly used antibiotics, driven by overprescription, self-medication, and inadequate regulatory measures. Key pathogens included multidrug-resistant *Klebsiella pneumoniae* and *Escherichia coli*. The authors stressed the urgent need for robust antimicrobial stewardship programmes, enhanced laboratory surveillance, and regional collaboration to combat AMR effectively. Addressing this issue is vital to safeguarding the efficacy of antibiotics and improving healthcare outcomes across the Caribbean [8].

Da Silva et al. reviewed the prevalence of methicillin-resistant *Staphylococcus aureus* (MRSA) in food and its occurrence in Brazil. The study highlighted contamination in various food products, particularly meat and dairy, posing a potential risk to public health. MRSA strains exhibited significant antimicrobial resistance, complicating treatment options. Factors contributing to contamination included poor hygiene during food processing and handling. The authors emphasised the need for stricter food safety regulations, improved hygiene practices, and enhanced surveillance to mitigate the spread of MRSA in food and reduce its associated health risks in Brazil [9].

Summary: The studies highlight escalating challenges of infections and antimicrobial resistance in the Caribbean and Latin America. Key issues include healthcare-associated infections, rising carbapenemase-producing organisms, inconsistent hand hygiene, and multidrug-resistant pathogens like *Klebsiella pneumoniae*, *Escherichia coli*, and MRSA. Urgent measures, including enhanced infection prevention, antimicrobial stewardship, and regional cooperation, are essential to mitigate these public health threats.

### 2.2. Methicillin-Resistant Staphylococcus aureus (MRSA)

#### 2.2.1. Introduction

Methicillin-resistant *Staphylococcus aureus* (MRSA) has been a global concern since its emergence in British hospitals in the 1960s [10]. The 1965 paper by Colley et al. examines the incidence of methicillin-resistant *Staphylococcus aureus* (MRSA) in a general British hospital. The study highlights the presence of methicillin-resistant staphylococcal strains, which were a growing concern at the time [11].

#### 2.2.2. Trinidad and Tobago

In Trinidad and Tobago, antibiotic resistance in *Staphylococcus aureus* poses a significant public health concern. Akpaka et al. (2016) identified a high prevalence of acquired resistant genes targeting *blaZ* in methicillin-resistant *Staphylococcus aureus* (MRSA), particularly associated with the ST239-MRSA III strain. Additionally, resistance genes such as *vanA*, *ermB*, and *cfr* were detected. Despite generally high susceptibility rates (>80%), trimethoprim/sulfamethoxazole emerged as the most effective antibiotic among 19 tested. Notably, *S. aureus* isolates from paediatric wards exhibited higher susceptibility rates compared to isolates from major hospitals [12]. Akpaka et al. (2017) further studied *Staphylococcus aureus* isolates from Trinidad and Tobago, reporting high resistance rates to antibiotics such as cloxacillin and ciprofloxacin; MRSA isolates also exhibited resistance to clindamycin and vancomycin. These findings underscore the necessity for enhanced infection control measures in healthcare settings to mitigate the burden of MRSA [12].

Another study investigated MRSA prevalence in broilers and workers at “pluck shops” in Trinidad. While MRSA was isolated from 0.7% of broilers, none was detected in workers. All MRSA isolates displayed resistance to multiple antimicrobial agents. The study marked the first detection of MRSA in Trinidadian poultry, raising public health concerns about potential occupational exposure and subsequent human infection risks. Rapid identification of MRSA strains carrying SCCmec genes remains crucial. Surveillance for resistance mechanisms in *S. aureus* should be prioritised, and the cost of detecting MRSA in clinical specimens should be included in routine laboratory testing to prevent mortality [13].

A study conducted in surgical wards of a public hospital in Northern Trinidad found a 39.5% MRSA prevalence. Identified risk factors for colonisation included patients aged 60–69 years, those with comorbidities, hospital stays exceeding one week, prior surgeries, and previous antibiotic use [14]. The study by Chroboczek et al. characterised MRSA clones in human populations across five West Indian islands, demonstrating that MRSA prevalence may vary across the Caribbean [14].

In summary, MRSA remains a significant public health challenge in Trinidad and Tobago, with varying prevalence and resistance patterns across healthcare settings and community environments. Effective strategies, including stringent hand hygiene, antibiotic stewardship, and molecular surveillance, are essential for managing MRSA transmission and reducing its clinical impact.

#### 2.2.3. Jamaica, West Indies

Antibiotic resistance (ABR) is a significant global health challenge, exacerbated by the declining production of new antibiotics. Consequences of ABR include increased mortality, prolonged hospital stays, and rising healthcare costs. Limited data on ABR are available in the English-speaking Caribbean, with notable research from Jamaica, Trinidad and Tobago, and Barbados. A cross-sectional study in Jamaica assessed the knowledge, attitudes, and practices of physicians regarding ABR. A 32-item questionnaire was administered to 800 doctors, representing 20% of the island’s physicians, between October 2014 and September 2015. The study aimed to inform the development of national antibiotic guidelines and educational initiatives. Among 695 respondents (87% response rate), 51% were female, with 60% practising in hospitals and 51% having postgraduate training. Most respondents recognised the global importance of ABR, though fewer acknowledged its local impact or relevance to their practice. Hospital-based physicians showed greater awareness of ABR compared to outpatient-based physicians. While the majority identified ceftriaxone and ciprofloxacin as resistance-inducing, amoxicillin–clavulanic acid was incorrectly perceived as highly resistance-inducing. Variability in the treatment duration for infections suggested the need for clearer guidelines. The study emphasised the importance of educational programmes, timely laboratory reports, and improved access to laboratory services to enhance ABR management. Addressing these issues could mitigate the impact of ABR in Jamaica and influence broader healthcare practices across the Caribbean [15].

Methicillin-resistant *Staphylococcus aureus* (MRSA) is a significant pathogen responsible for both hospital-acquired (HA) and community-acquired (CA) infections, leading to high morbidity, mortality, and healthcare costs. CA-MRSA, which differs from HA-MRSA in genetic composition and resistance profile, is increasingly prevalent in environments with close physical contact. In the Caribbean, including Jamaica and Trinidad and Tobago, the prevalence of MRSA is rising, with distinct clones circulating. A study in public hospitals in Kingston and St. Andrew, Jamaica, analysed antimicrobial susceptibility, resistance mechanisms, and the genetic typing of 61 MRSA isolates between September 2011 and August 2012. High resistance rates were observed for ciprofloxacin (86.9%), erythromycin (85.3%), lincomycin (80%), and clindamycin (74%). No resistance was detected for vancomycin and teicoplanin. Mupirocin resistance included 23% low-level resistance and 18% high-level resistance. Gentamicin resistance was observed in 64% of isolates. SCCmec type IV was the most common (85%), followed by type II (9%). Sixteen distinct multiple-locus variable number tandem repeat (MLVA) patterns were identified, with pattern 10 being the most prevalent. These findings highlight the emergence of new MRSA lineages in Jamaica and underscore the need for accurate susceptibility testing, continuous surveillance, and tailored treatment strategies to manage MRSA infections effectively [16].

Monecke et al. (2023) investigated the distribution of *Staphylococcus aureus* clonal complexes across the Caribbean Islands, providing a detailed analysis of the genetic diversity and epidemiology of this pathogen. Using multi-locus sequence typing (MLST), the study classified isolates into various clonal complexes (CCs), revealing a rich diversity, including globally recognised types such as CC5 and CC8 alongside unique Caribbean variants. The findings contribute to understanding the regional epidemiology of *S. aureus* and highlight the need for collaborative efforts in monitoring and controlling antibiotic resistance within the Caribbean region. Such efforts are critical to improving clinical outcomes and guiding effective public health strategies [17].

The study highlights the interplay between local and global *Staphylococcus aureus* strains, linking clonal complexes to antibiotic resistance profiles that influence treatment options and patient outcomes. It emphasises the need for region-specific surveillance to address emerging resistance patterns and adapt treatment strategies. Mapping clonal complex distribution and tracking the changes in prevalence and resistance are critical for effective infection control and tailored public health interventions in the Caribbean. Monecke et al.’s findings provide valuable insights into *S. aureus* epidemiology, underscoring the importance of continuous monitoring, targeted infection control, and adaptive health strategies to manage and prevent infections in the region [17].

#### 2.2.4. Cuba

Leiva Peláez et al. (2015) conducted a study on the molecular epidemiology of MRSA across four Cuban hospitals, aiming to characterise the prevalent strains and their genetic profiles. The study revealed a diverse array of MRSA strains, including locally and globally disseminated clones. Resistance to methicillin was high, posing significant treatment challenges. The presence of key resistance genes and hospital-specific clonal complexes indicated the potential for outbreaks in healthcare settings. The authors emphasised the importance of monitoring MRSA strains to track resistance and implement effective infection control measures. The study provides crucial insights into MRSA epidemiology in Cuba, reinforcing the need for ongoing surveillance and targeted interventions to manage MRSA-related infections [18].

#### 2.2.5. Puerto Rico

A pilot study by Marrero Rolón et al. (2016), prompted by Puerto Rico’s Law #298, evaluated a pre-operative MRSA decolonisation protocol at Manatí Medical Center for elective orthopaedic surgeries. The protocol included mupirocin nasal ointment and chlorhexidine washes, with pre-admission nasal swabs tested via PCR. High colonisation rates of MRSA/MSSA were detected but insufficient time for decolonisation was a potential barrier. Patient education was crucial for compliance. The protocol significantly reduced MRSA infections among patients who adhered to it, demonstrating practical benefits such as enhanced infection control and improved outcomes. The study recommends routine pre-admission screening and decolonisation to minimise infection risks and align with legislative mandates, contributing to improved patient safety and quality of care [19].

Rochet et al. (2020) characterised pathogens isolated from cutaneous abscesses in patients evaluated by the Dermatology Service at a Puerto Rican emergency department. The study identified *Staphylococcus aureus*, predominantly MRSA (85%), as the leading pathogen, followed by various Gram-negative bacteria and anaerobes. Incision and drainage were performed in all cases, with oral clindamycin being the most common systemic therapy (36%). While tetracycline and vancomycin were effective against all MRSA strains, 14.3% of isolates showed clindamycin resistance. These findings highlight the high prevalence of MRSA in emergency department patients and emphasise the need for comprehensive pathogen identification and susceptibility testing to guide treatment decisions. In addition, the study underscores the importance of monitoring clindamycin resistance and reconsidering systemic antibiotic use to manage infections effectively. The results contribute valuable data for improving infection management and antibiotic stewardship in emergency dermatology settings [20].

In summary, studies from Cuba and Puerto Rico reveal critical insights into the epidemiology of MRSA and its resistance patterns in the Caribbean. In Cuba, investigations highlight the emergence of MRSA in community settings, the presence of the highly virulent USA300 strain, and the genetic diversity of MRSA in hospitals. These findings underscore the need for enhanced surveillance, infection control, and public health strategies to address resistant pathogens. In Puerto Rico, research emphasises the significance of addressing underlying chronic conditions, implementing pre-operative decolonisation protocols, and tailoring antibiotic therapies based on resistance patterns. The rising prevalence of MRSA and its associated challenges call for coordinated efforts to manage and mitigate its impact on public health in the region. Strengthened surveillance, antimicrobial stewardship, and international collaboration are crucial for controlling the spread of MRSA and improving patient outcomes in the Caribbean.

#### 2.2.6. Dominican Republic

A retrospective study conducted in three tertiary hospitals in Santiago de los Caballeros, Dominican Republic, analysed 3802 clinical microbiology reports from 2016 to 2017. Methicillin resistance was prevalent in *Staphylococcus aureus* (57.3%). This finding emphasises the urgent need for measures to mitigate antimicrobial resistance and its adverse clinical implications in the Dominican Republic [21].

#### 2.2.7. Haiti

A cross-sectional study in Haiti revealed low carriage rates of *Staphylococcus aureus* (8.4% MSSA and 2.8% MRSA), with high tetracycline resistance among isolates. MSSA isolates carried Panton–Valentine leukocidin (PVL) and toxic shock syndrome toxin (TSST) genes, whereas MRSA lacked virulence markers. Unique MSSA phenotypes, including linezolid-resistant strains, were identified. PVL-positive MSSA isolates showed expansion of the ST152 clone, previously limited to Africa. These findings highlight the importance of understanding *S. aureus* epidemiology to inform effective treatment and public health strategies [22].

#### 2.2.8. St. Kitts and Nevis

In St. Kitts and Nevis, a study on antimicrobial resistance found that 45% of 152 *Staphylococcus aureus* isolates were MRSA. High resistance rates were observed for erythromycin (55%), moxifloxacin (41%), and levofloxacin (40%), while susceptibility to ceftaroline, linezolid, teicoplanin, telavancin, and vancomycin was retained. Whole genome sequencing showed that 88% of MRSA isolates belonged to ST8, with genetic diversity within the USA300 North American Epidemic lineage. All ST8 strains clustered within the PVL-positive, ACME-positive, and SCCmec IVa subtypes. These findings highlight significant methicillin, macrolide, and fluoroquinolone resistance, calling for enhanced surveillance and antimicrobial stewardship to manage resistance effectively [23].

#### 2.2.9. Guyana

In Guyana, a study of skin and soft-tissue-infection isolates from an emergency department found a high frequency (51%) of MRSA. All strains were SCCmec type IV, PVL-positive, and LukAB-positive, resembling the USA300 epidemic clone. These findings underscore the importance of tailored antibiotic therapy and infection control measures to manage MRSA in regions with similar epidemiological characteristics [24].

#### 2.2.10. Brazil

In another study, 116 MRSA isolates from 12 cystic fibrosis (CF) paediatric patients in Rio de Janeiro were examined over six years. The study found high resistance to erythromycin and ciprofloxacin, with SCCmec IV being the most common type. Pulsed-field gel electrophoresis revealed genetic diversity, though certain spa types persisted within patients, indicating lineage persistence in chronic lung infections. These findings highlight the challenges posed by MRSA in CF patients and the need for tailored management strategies [25].

The CC5-II group, prevalent in Central and North America, has also been found in South American nations such as Brazil, Venezuela, and Ecuador. This group includes isolates carrying the CC5-SCCmecII element, linked to the USA100 lineage and the NY clone. The study reveals polyphyletic evolutionary pathways for these MRSA isolates, demonstrating the need for molecular surveillance to understand their development and spread [26].

#### 2.2.11. Community-Acquired *Staphylococcus aureus* ST8 in the Caribbean, South America, and Globally

Community-acquired *Staphylococcus aureus* (CA-MRSA) ST8 is a significant pathogen with global public health implications. This lineage, including the hypervirulent USA300 clone, is notable for its rapid dissemination and ability to cause severe infections outside healthcare settings. Its adaptation to community environments highlights its evolutionary success and capacity to thrive in diverse settings.

In the Caribbean and South America, CA-MRSA ST8 has emerged as a pressing health concern. Studies reveal its prevalence among outpatients in the Caribbean, reflecting its persistence and adaptability in community settings. Similarly, evidence from South America, including countries such as Brazil and Argentina, indicates widespread ST8 transmission. Factors such as high rates of skin infections, crowded living conditions, and inconsistent healthcare access contribute to its spread. The clone’s antibiotic resistance further exacerbates its impact on public health.

Globally, the transmission of ST8 underscores the importance of understanding its epidemiology and implementing effective control measures. Surveillance and molecular characterisation of MRSA strains in the Caribbean and South America are essential for tracking the spread and evolution of this lineage. Public health strategies should prioritise improved infection control practices, enhanced hygiene, and better access to healthcare to mitigate the effects of ST8.

Strauß et al. (2017) used comparative genomics to study CA-MRSA ST8, including USA300, analysing 224 isolates. They traced the lineage’s emergence to Central Europe in the mid-19th century and its export to North America in the early 20th century. The lineage acquired traits such as Panton–Valentine leukocidin (PVL), SCCmec IVa, ACME, and a capsular polysaccharide gene mutation. While the PVL-encoding phage was introduced once, SCCmec types were acquired at different times and locations. The study highlights the origin, evolution, and global dissemination of this hypervirulent lineage [27].

Two phylogenies have confirmed that USA300 is divided into two distinct lineages, USA300-NAE and USA300-SAE, which share a common ancestor from approximately 50 years ago. Isolates with gene combinations characteristic of USA300-NAE and USA300-SAE, such as ACME, COMER, and cap5E mutations were exclusive to these clades, confirming their distinctiveness within the ST8 lineage [27].

CC8, a major *Staphylococcus aureus* lineage, has produced significant MRSA clones, including USA300. Since 2000, USA300 has dominated community infections in the United States, marked by features such as cap5D/E mutations, SaPI5, PVL, SCCmecIVa, and ACME [28].

In Europe, the ST80-IV “European” clone, originating from West Africa, remains prevalent in some North African countries but has declined in Europe. Sporadic occurrences of USA300-NA MRSA in Europe, particularly in Geneva, Switzerland, are linked to importations from the Americas [29].

### 2.3. Extended-Spectrum Beta-Lactamases (ESBLs)

#### 2.3.1. Introduction

Extended-spectrum beta-lactamases (ESBLs) are enzymes produced by certain bacteria, predominantly within the Enterobacteriaceae family, that confer resistance to a broad range of beta-lactam antibiotics. These include extended-spectrum cephalosporins such as cefotaxime, ceftazidime, and aztreonam, as well as monobactams like aztreonam. Encoded by genes located on mobile genetic elements such as plasmids, ESBLs can transfer between bacterial strains and species, facilitating their dissemination.

The prevalence of ESBL-producing bacteria has risen globally, particularly in healthcare settings where selective pressure from extensive antibiotic use is high. These bacteria contribute to healthcare-associated infections, including urinary tract infections, bloodstream infections, and pneumonia, which are challenging to treat due to limited antibiotic options. Effective management requires rapid pathogen identification, antimicrobial susceptibility testing, and infection control measures, such as stringent hand hygiene, environmental cleaning, and antimicrobial stewardship programmes. Continued surveillance and research are essential to address the threat posed by ESBLs, preserve antibiotic efficacy, and develop novel treatment strategies [30].

#### 2.3.2. Definition

Beta-lactamases are enzymes that hydrolyse beta-lactam antibiotics, rendering them ineffective. Extended-spectrum beta-lactamases (ESBLs), a concerning subset, inactivate a wide range of antibiotics, including first-, second-, and third-generation cephalosporins, penicillins, and monobactams, although they remain susceptible to clavulanic acid inhibition. Beta-lactamases are classified using functional (Bush–Jacoby–Medeiros) and molecular (Ambler) schemes [31].

#### 2.3.3. Detection of ESBLs in Medical Institutions

In Mexico, ESBL-producing *Escherichia coli* and *Klebsiella pneumoniae* were detected in isolates from healthcare institutions in Hermosillo, Sonora, with prevalence rates of 11.9% and 8.7%, respectively. High resistance to ciprofloxacin (88%), trimethoprim/sulfamethoxazole (72%), and aminoglycosides (59%) was observed, while meropenem, amikacin, and tigecycline remained effective. The predominant ESBL variants were CTX-M-1 (88%) and CTX-M-9 (5%). Genetic diversity was noted among hospital and community isolates, highlighting the need for ongoing epidemiological surveillance [32].

In Brazil, bloodstream infections in ICUs necessitate monitoring for ESBL-resistant bacteria. A study from the University Hospital’s ICUs identified *Klebsiella pneumoniae* and *Staphylococcus aureus* among the isolates, many exhibiting carbapenem resistance. This highlights the need for enhanced infection control and targeted antibiotic strategies [33].

Wildlife has also been implicated in antibacterial-resistance spread. In Guadeloupe, *E. coli* from wild animals exhibited low antibiotic resistance rates (5.4%) but the blaCTX-M-1 gene, shared with humans and birds, was identified on IncI1 plasmids. While ESBL-producing *E. coli* were rare, the widespread distribution of blaCTX-M-1 plasmids underscores the need for surveillance to mitigate interspecies transmission risks [34].

In Trinidad and Tobago, a study of 66 *Klebsiella pneumoniae* isolates found ESBL production in 78.8%, predominantly carrying blaSHV (84.8%) and blaCTX-M (46.9%) genes. Some isolates showed epidemiological links, indicating local dissemination. These findings underscore the necessity of robust molecular characterisation and surveillance to combat ESBL resistance in the Caribbean [35].

In Haiti, a point-prevalence survey at an obstetrics emergency hospital revealed high ESBL colonisation rates in neonates (51%) compared to mothers (10%). Risk factors included prolonged hospitalisation and antibiotic use, highlighting the importance of stringent infection control measures [36].

In Guadeloupe, a study of the *Enterobacter cloacae* complex (ECC) identified 57 ESBL-producing strains among 94 third-generation cephalosporin-resistant isolates. Whole-genome sequencing revealed the blaCTXM-15 gene on conserved plasmids, with limited genetic diversity among strains. Higher nosocomial ESBL infections in Guadeloupe compared to mainland France underscore the need for effective management and surveillance [37].

A study in Trinidad examined 373 *Klebsiella pneumoniae* isolates, identifying high ESBL activity with blaTEM, blaSHV, and blaCTX-M genes, alongside carbapenem resistance genes blaKPC and blaNDM-1. Virulence factor genes were also prevalent. This comprehensive study highlights the importance of integrating molecular techniques into diagnostics to better understand resistance mechanisms [38].

At Guadeloupe University Hospital, an ICU stewardship programme with restrictive antibiotic policies led to reduced ESBL incidence, lower antibiotic use, and improved patient outcomes. This demonstrates the feasibility and effectiveness of tailored stewardship programmes in combating antimicrobial resistance in critical care settings [39].

#### 2.3.4. Detection of ESBLs in the Environment

Antibiotic-resistance dissemination from wastewater treatment plants to terrestrial animals poses a significant public health concern. A study identified extended-spectrum β-lactamase (ESBL)-producing *Enterobacter cloacae* complex strains in clinical and environmental samples, revealing the prevalence of ST114 across human and nonhuman hosts. The presence of a common IncHI2/ST1/blaCTX-M-15 plasmid highlights the potential for global spread, emphasising the need for surveillance and intervention strategies [40].

Another study in Guadeloupe investigated *Escherichia coli* resistance in recreational freshwater sites. Although faecal contamination was evident, no ESBL-producing *E. coli* strains were identified. Resistance to streptomycin and tetracycline was observed in some isolates. Biofilm analysis revealed clinically relevant resistance genes, though overall antibiotic resistance levels in recreational waters were low [41].

Research on the *Klebsiella pneumoniae* species complex (KpSC) in Guadeloupe analysed 590 samples, finding ESBL production in 21.4% of isolates, primarily from hospitals. High-risk multidrug-resistant (MDR) clones such as ST17, ST307, and ST11 were prevalent, though human-to-nonhuman transmission was limited. Genomic surveillance is crucial for containing the spread of MDR pathogens [42].

In pets in Guadeloupe, 7.6% of rectal swabs from dogs and cats carried ESBL-producing Enterobacteriaceae, with shelter stays being a significant risk factor. Most isolates were *E. coli* carrying blaCTX-M-1/IncI1-Iγ/ST3 plasmids, underscoring the need for improved hygiene practices among pet owners [43].

#### 2.3.5. Drug-Resistant Bacteria in the Region

A Caribbean-wide survey revealed high usage of β-lactam antibiotics, including cephalosporins, quinolones, and macrolides. Despite funding cessation for CARPHA-based AMR surveillance, phenotypic and genomic analyses provided insights into resistance determinants and the phylogenetic distribution of *Klebsiella pneumoniae* in the region [44].

Antimicrobial resistance (AMR) remains a significant global health threat. In the Americas in 2019, bacterial AMR was associated with 569,000 deaths and 141,000 directly attributable fatalities. Lower respiratory infections accounted for the largest AMR burden, followed by bloodstream and intra-abdominal infections. Six key pathogens, including *Staphylococcus aureus*, *Escherichia coli*, and *Klebsiella pneumoniae*, were responsible for most AMR-related deaths. These findings highlight the need for tailored policy interventions and multisectoral collaboration to address AMR effectively [45].

The World Health Organization (WHO) Region of the Americas reported 920,000 bacterial infection deaths in 2019, with 38% linked to AMR. Lower respiratory, bloodstream, and intra-abdominal infections dominated, underscoring the critical role of resistance in worsening infection outcomes. Effective interventions are needed to mitigate the public health impact of AMR [46].

A “one health” approach in Guadeloupe assessed *Enterobacter cloacae* complex (ECC) strains from human, animal, and environmental sources. Third-generation cephalosporin resistance in nonhuman samples (16.2%) was linked to AmpC overproduction, while human samples predominantly exhibited ESBL production. No inter-compartmental circulation was observed but environmental factors likely contribute to resistance in nonhuman isolates, warranting further investigation [47].

## 3. Epidemiology of Infectious Diseases in the Caribbean and South America

The epidemiology of nosocomial infections (HAIs) in the Caribbean and Latin America reflects gaps in the healthcare infrastructure and a high burden of infectious and non-infectious conditions. Addressing these issues requires strengthened infection control measures, enhanced screening protocols, and improved diagnostics, especially for immunodeficiencies and autoimmune disorders. Bardach et al. (2024) conducted a systematic review and meta-analysis on invasive pneumococcal disease (IPD) in Latin America and the Caribbean, focusing on serotype distribution, disease burden, and vaccination impact. The study highlighted the substantial burden of IPD, with serotype variations across regions. Vaccination programmes, particularly those using pneumococcal conjugate vaccines, significantly reduced disease incidence and altered serotype prevalence. However, non-vaccine serotypes emerged, requiring ongoing surveillance and adaptation of vaccination strategies. The findings underscore the importance of sustained immunisation efforts to mitigate the public health impact of IPD in the region [48].

The International Nosocomial Infection Control Consortium (INICC) identified risk factors, such as prolonged mechanical ventilation, central line use, and extended catheterisation, as critical contributors to mortality in ICUs across 12 Latin American countries over 24 years. Infections from pathogens like *Salmonella* spp., *Toxoplasma gondii*, *Leishmania* spp., and *Plasmodium* spp. pose additional risks in regions with inadequate infection control [49].

Primary immunodeficiency and autoimmune disorders further increase susceptibility to HAIs, underscoring the need for tailored infection prevention strategies for vulnerable populations [50]. Enhanced infection control, surveillance, and interventions are essential to mitigate the HAI burden in the region.

Microbiome studies in Latin America and the Caribbean reveal its role in health, disease, and environmental interactions. Factors like diet, infectious diseases, and urbanisation shape regional microbiome profiles, influencing malnutrition, chronic diseases, and antimicrobial resistance. Expanding microbiome research could inform public health interventions and sustainable agricultural practices [51].

## 4. Salmonellosis

The global pandemic of *Salmonella* infections, first linked to contaminated eggs in the late 1980s, remains a major public health concern. Contamination sources include chicks, feed, rodents, and farm environments, necessitating improved biosecurity and waste management practices. Denmark’s success in reducing *Salmonella* through treated animal feed underscores the potential of targeted interventions [52].

An immunoepidemiological study assessed anti-*Salmonella* antibodies in humans and chickens. Results showed 11.3% seropositivity in human samples and 95.3% in chicken IgY samples, indicating endemic *Salmonella* in poultry. The findings call for further research into transmission dynamics and implications for public health [53].

The cross-sectional study characterised *Salmonella* isolates from hatcheries, broiler farms, processing plants, and retail outlets in Trinidad and Tobago using whole-genome sequencing (WGS). Twenty-three serovars were identified, with Kentucky (20.5%), Javiana (19.2%), Infantis (13.7%), and Albany (8.9%) being predominant. WGS showed 76% agreement with traditional serotyping. Antimicrobial resistance genes for aminoglycosides, cephalosporins, and other antimicrobials were detected, with 6.8% of isolates showing multidrug resistance (MDR), all from broiler farms. Virulence factors associated with secretion systems (69.3%) and fimbrial adherence (29.2%) were prevalent. Notably, 50% of *Salmonella Infantis* MDR isolates contained the blaCTX-M-65 gene. The study provides critical genomic data for epidemiological tracking of *Salmonella* outbreaks in the Caribbean and beyond [54].

The long-term impact of antibiotic use, such as ceftiofur, on *Salmonella* resistance patterns in the poultry and dairy industries must be addressed. Prolonged exposure can select for resistant strains, complicating control efforts and posing significant public health risks. Sustainable antibiotic practices are essential to preserving treatment efficacy for both human and animal infections [55].

## 5. The Risk of Transfusion-Transmitted Infections

Blood transfusions are critical in medical treatments, saving lives during surgery, injury, or illness; however, they carry risks, particularly the transmission of bacterial, viral, and parasitic pathogens. Among these, transfusion-transmitted bacterial infections (TTBIs) pose a notably higher risk than viral infections, especially in platelet transfusions, which are prone to bacterial contamination due to room-temperature storage. Ensuring blood safety remains a global healthcare priority [56,57].

Viral pathogens transmitted via blood transfusions include HIV, hepatitis C (HCV), hepatitis B (HBV), hepatitis A (HAV), West Nile virus (WNV), cytomegalovirus (CMV), SARS-CoV-1, human T cell lymphotropic viruses (HTLVs), Zika virus, and parvovirus B19. These pose significant risks, particularly for immunocompromised patients, potentially leading to severe health complications. While nucleic acid amplification tests (NAATs) have markedly reduced viral infections in transfusions, bacterial infections remain challenging. Stored blood products, especially platelets, provide an ideal environment for bacterial growth, necessitating stringent detection and prevention strategies [58,59].

In the Caribbean and Latin America, endemic diseases such as toxoplasmosis, leishmaniasis, and malaria pose significant transfusion risks. *Toxoplasma gondii*, which causes toxoplasmosis, can be transmitted through infected blood products, particularly to immunocompromised patients. Similarly, *Leishmania* spp. have been documented in transfusion scenarios, with visceral leishmaniasis presenting risks from asymptomatic donors. *Plasmodium* spp., responsible for malaria, present a major concern in endemic regions, with asymptomatic carriers posing challenges for safe blood donation. Efforts to mitigate these risks include enhanced diagnostics, stricter donor deferral policies, and robust screening programmes [60,61].

### Microorganisms Transmitted Through Blood Transfusion

Blood transfusion carries the risk of transmitting various microorganisms, including bacterial, viral, parasitic, and prion-related pathogens. Common bacterial pathogens include *Yersinia enterocolitica*, *Pseudomonas fluorescens*, *Enterobacter* spp., *Serratia* spp., *Escherichia coli*, *Streptococcus viridans*, *Streptococcus bovis*, beta-haemolytic streptococci, coagulase-negative staphylococci, *Staphylococcus aureus*, as well as *Treponema pallidum*, which causes syphilis. Viral risks involve hepatitis viruses (HAV, HBV, and HCV), HIV-1 and HIV-2, human T-lymphotropic virus (HTLV-I/II), West Nile virus (WNV), cytomegalovirus (CMV), Zika virus, SARS-CoV-1, and parvovirus B19. Parasitic pathogens such as *Plasmodium* spp. (malaria), *Trypanosoma cruzi* (Chagas disease), *Babesia microti* (babesiosis), and *Leishmania* spp. pose additional risks, alongside prion diseases like variant Creutzfeldt–Jakob disease (vCJD). Key concerns include bacterial contamination in platelets stored at room temperature, viral transmission despite rigorous screening protocols, and parasite transmission from asymptomatic donors. Continuous advancements in testing and safety measures remain critical to minimising these risks and ensuring the safety of blood transfusions [62].

## 6. Dengue in the Caribbean and Latin America

Dengue fever, caused by the dengue virus and transmitted primarily by *Aedes aegypti* mosquitoes, remains a major public health concern in the Caribbean and Latin America. The tropical climate and high urban population densities provide ideal conditions for mosquito proliferation. Cyclical epidemics, often linked to climatic changes such as El Niño, exacerbate the disease burden, straining healthcare systems with hospital admissions and fatalities from dengue haemorrhagic fever or dengue shock syndrome [63].

Preventive measures, including vector control programmes and public awareness campaigns, are widely implemented but face challenges such as inadequate funding and public compliance. Advancements in vaccine development, such as the Dengvaxia vaccine, offer hope, though issues like variable efficacy across serotypes, accessibility, and public hesitancy persist. Socio-economic factors, including poor sanitation and inadequate housing, exacerbate mosquito breeding, underscoring the need for multifaceted approaches combining surveillance, infrastructure strengthening, and community engagement [64,65].

## 7. Confronting Tuberculosis and HIV Screening in Resource-Limited Settings

### 7.1. Tuberculosis and HIV Screening

Addressing tuberculosis (TB) and HIV screening in resource-limited settings presents ethical and practical challenges requiring targeted strategies for effective control. Stigma and discrimination deter participation, perpetuating transmission and delaying treatment. Ensuring informed consent and confidentiality is challenging in settings with low literacy and inadequate healthcare infrastructure, raising concerns about patient autonomy and trust. Unequal access to screening disproportionately affects vulnerable populations, including rural communities and migrants, highlighting equity issues. Mandatory screening may conflict with voluntariness principles, exacerbating mistrust. Limited infrastructure, insufficient laboratory facilities, and funding constraints hinder accurate and timely screening, particularly for HIV. Integrating TB and HIV services is logistically complex, and stock-outs of diagnostic kits erode public trust. Cultural misconceptions, especially regarding HIV, further impede acceptance. Multifaceted interventions, such as community engagement, task-shifting to community health workers, and investments in portable diagnostics, are essential. Global partnerships and sustainable funding are vital to scaling up screening programmes and improving health outcomes [66].

### 7.2. Mincle, a Macrophage Receptor Involved in Mycobacterial Recognition

Mincle, a macrophage receptor involved in mycobacterial recognition, plays a crucial role in TB immunity. Computational tools have advanced understanding of Mincle’s interactions with mycobacterial ligands like trehalose dimycolate, opening avenues for therapeutic research. While promising, limitations in computational models necessitate further investigation to optimise applications for TB management. Figure 1 shows the five most feasible poses in which trehalose dibenzenate, the derivative analogue of trehalose dimycolate, binds to mincle [67].

## 8. Infections in the Course of Immunological Disorders

Espinoza Mora et al. (2022) conducted a cross-sectional study to examine primary immunodeficiency diseases (PIDs) among Costa Rican adults, revealing significant underdiagnosis and delayed identification. The study involved 200 individuals with recurrent infections and immune dysfunction, identifying common variable immunodeficiency (CVID) and selective IgA deficiency as the most prevalent conditions. These PIDs were linked to increased susceptibility to respiratory and gastrointestinal infections, highlighting the clinical burden on affected individuals. The authors emphasised the need for heightened awareness, improved diagnostic protocols, and targeted interventions to enhance PID detection and management in Costa Rica. Their findings contribute to understanding the regional epidemiology of PIDs and underscore the importance of addressing these conditions to improve patient outcomes and healthcare strategies in Central America [68]

In the study by Espinoza Mora et al. (2022), the microorganisms identified in association with primary immunodeficiency diseases (PIDs) in Costa Rican adults primarily included pathogens responsible for recurrent respiratory and gastrointestinal infections. These microorganisms [68] are outlined below:Bacteria:-*Streptococcus pneumoniae*: commonly causing pneumonia and sinusitis;-*Haemophilus influenzae*: linked to respiratory tract infections;-*Staphylococcus aureus*: found in cases of skin and soft tissue infections.Viruses:-Cytomegalovirus (CMV): associated with chronic infections in immunocompromised individuals;-Epstein–Barr virus (EBV): linked to lymphoproliferative disorders in PIDs.Fungi:-*Candida albicans*: causing mucosal infections like oral thrush and oesophagitis;-*Aspergillus* species: associated with chronic pulmonary infections.Parasites:-*Giardia lamblia*: responsible for persistent diarrhoea in immunocompromised patients.

The study highlights the susceptibility of PID patients to a broad spectrum of pathogens, underscoring the need for vigilant infection management.

### 8.1. Severe Combined Immunodeficiency Disorders (SCIDs)

Severe combined immunodeficiency disorders (SCIDs) result in profound T and B cell deficiencies, leaving patients highly vulnerable to infections caused by common and rare microorganisms. Bacterial infections frequently include *Staphylococcus aureus*, *Pseudomonas aeruginosa*, and *Streptococcus pneumoniae*, with rare pathogens like *Nocardia* and *Mycobacterium avium* complex also observed. Viral infections are common, with *cytomegalovirus* (CMV), *Epstein–Barr virus* (EBV), and adenovirus being prevalent, alongside rare but severe complications from measles and certain enteroviruses. Fungal threats include *Candida* species, *Aspergillus*, and occasionally *Cryptococcus neoformans* and *Pneumocystis jirovecii*. Parasitic infections such as *Toxoplasma gondii* and *Cryptosporidium*, as well as rare parasites like *Strongyloides stercoralis*, can cause severe complications. Prompt diagnosis and treatment are critical as even relatively harmless organisms can lead to severe or disseminated infections in SCID patients [69].

### 8.2. Chronic Granulomatous Disease (CGD)

Chronic granulomatous disease (CGD) is a primary immunodeficiency caused by defects in phagocyte function, impairing the production of reactive oxygen species required to kill pathogens. Patients are susceptible to recurrent and severe infections by *Staphylococcus aureus*, *Serratia marcescens*, *Burkholderia cepacia*, *Nocardia*, and *Salmonella*. Fungal infections, particularly those caused by *Aspergillus* and *Candida* species, are common and often life-threatening. Chronic infections and granulomas typically affect the lungs, liver, and lymph nodes. Lifelong antimicrobial prophylaxis and, in some cases, stem cell transplantation are required for effective management [70].

### 8.3. Transient Hypogammaglobulinemia of Infancy (THI)

Transient hypogammaglobulinemia of infancy (THI) is characterised by low levels of immunoglobulin G (IgG), leading to recurrent respiratory tract infections such as sinusitis, otitis media, and bronchitis caused by *Haemophilus influenzae*, *Streptococcus pneumoniae*, and *Moraxella catarrhalis*. Viral infections, including respiratory syncytial virus (RSV) and adenovirus, can complicate the condition. Gastrointestinal infections occur less frequently. Severe or life-threatening infections are rare, and most infants outgrow the condition by age 2 to 4. Management often involves treating active infections with antibiotics, with prophylactic antibiotics or immunoglobulin replacement therapy used in select cases [71,72].

### 8.4. Neuropsychiatric Systemic Lupus Erythematosus (NPSLE)

Neuropsychiatric systemic lupus erythematosus (NPSLE) involves central and peripheral nervous system complications in systemic lupus erythematosus (SLE), manifesting as cognitive dysfunction, mood disorders, seizures, psychosis, and strokes. These patients face increased infection risks due to immunosuppressive therapy, immune dysfunction, and central nervous system involvement. Common infections include *Staphylococcus aureus*, *Escherichia coli*, and *Pneumocystis jirovecii*, as well as viral infections like herpes simplex virus (HSV) and CMV. Infections can mimic or exacerbate neuropsychiatric symptoms, complicating diagnosis. Effective management requires careful differentiation between lupus flares and infections. SLE is driven by immune complex formation, inflammatory cytokines, and complement activation, causing multisystem damage such as nephritis, arthritis, and vasculitis [73,74,75], as shown in Figure 2.

### 8.5. Interleukin-2 Receptor Alpha (IL-2Rα) Deficiency

Interleukin-2 receptor alpha (IL-2Rα) deficiency, a severe SCID variant, impairs regulatory T cell function, resulting in autoimmunity and recurrent infections. Patients frequently encounter *Staphylococcus aureus*, *Streptococcus pneumoniae*, CMV, EBV, and *Candida*. Haematopoietic stem cell transplantation is the only curative option. Patients with IL-2Rα deficiency often suffer from persistent viral and fungal infections due to severe immune dysregulation. Secondary complications, such as autoimmune diseases and chronic inflammatory disorders, further worsen the prognosis. Advances in genetic and cellular therapies offer hope for novel treatment modalities but haematopoietic stem cell transplantation remains the gold standard for restoring immune function. Close monitoring of infection markers and early intervention are essential to improve outcomes [76].

### 8.6. Multiple Sclerosis (MS)

Multiple sclerosis (MS) treatments often weaken the immune system, increasing susceptibility to respiratory infections like *Streptococcus pneumoniae* and *Haemophilus influenzae*, as well as UTIs and viral pathogens such as HSV and VZV. Certain infections, like EBV, may contribute to MS pathogenesis. Monitoring and early treatment of infections are critical. The role of the Epstein–Barr virus in MS pathogenesis has drawn significant research attention, suggesting potential links to immune dysregulation. Immunotherapy, while effective for MS symptoms, necessitates vigilance for opportunistic infections, particularly in patients receiving disease-modifying treatments. Comprehensive patient education and preventive vaccination programmes are pivotal in mitigating infection risks [77,78].

### 8.7. Ataxia–Telangiectasia (A-T)

Ataxia–telangiectasia (A-T) results in neurological dysfunction and immunodeficiency, leading to recurrent respiratory infections caused by *Haemophilus influenzae*, *Streptococcus pneumoniae*, and *Staphylococcus aureus*. Viral infections, such as RSV, can exacerbate symptoms. Management includes antibiotics and immunoglobulin replacement therapy. Patients with A-T often experience progressive respiratory compromise due to repeated infections and underlying structural lung changes. Early implementation of supportive respiratory care and regular surveillance for pulmonary function can significantly enhance quality of life. Advances in genetic therapies offer future potential for addressing the root causes of this disorder [79].

### 8.8. Allergic Bronchial Asthma

Allergic asthma is linked to airway inflammation often exacerbated by respiratory viruses (*rhinovirus* and *influenza*) and bacteria (*Haemophilus influenzae*). Early-life microbial exposure may modulate asthma development through immune regulation (hygiene hypothesis). Environmental modifications and allergen avoidance strategies remain central to asthma management. The role of airway microbiota in modulating immune responses is an emerging area of research, offering insights into personalised therapeutic approaches [80].

### 8.9. Leukocyte Adhesion Deficiency (LAD)

Leukocyte adhesion deficiency (LAD) impairs immune responses, resulting in severe bacterial infections (*Staphylococcus aureus* and *E. coli*), fungal pathogens (*Candida*), and delayed wound healing. Management focuses on infection prevention and haematopoietic stem cell transplantation in severe cases. Recent advancements in gene therapy show potential in addressing LAD’s underlying genetic defects. Enhanced understanding of molecular pathways has opened avenues for targeted treatments, complementing traditional antimicrobial strategies [81].

### 8.10. Chediak–Higashi Syndrome (CHS)

Chediak–Higashi Syndrome (CHS) involves defective lysosomal trafficking, leading to recurrent infections from *Staphylococcus aureus*, *Salmonella*, and *Candida*. EBV is linked to CHS’s accelerated phase. Haematopoietic stem cell transplantation is the only curative treatment. Comprehensive care, including regular monitoring for EBV-associated complications and supportive therapies for bleeding diathesis, is essential for CHS management. Experimental treatments focusing on lysosomal trafficking pathways hold promise for future interventions [82].

### 8.11. Myeloperoxidase (MPO) Deficiency

Myeloperoxidase (MPO) deficiency impairs neutrophil microbial killing, predisposing individuals to *Candida*, *Aspergillus*, and bacterial pathogens (*E. coli* and *Klebsiella*). Awareness and prompt treatment prevent severe complications. The asymptomatic nature of many MPO-deficient cases highlights the importance of routine screenings in high-risk populations. Ongoing research into oxidative burst augmentation therapies offers potential adjunctive strategies for infection control [83,84].

### 8.12. DiGeorge Syndrome

DiGeorge syndrome causes T cell deficiencies, increasing vulnerability to respiratory infections (*Streptococcus pneumoniae* and *Haemophilus influenzae*) and opportunistic pathogens (*Candida* and CMV). Early interventions and improved care enhance quality of life. Multidisciplinary approaches integrating genetic counselling, cardiac care, and immune monitoring are pivotal for optimising outcomes. Advances in thymus transplantation provide hope for addressing severe immunodeficiencies in DiGeorge syndrome [85,86].

### 8.13. Super-IgM Syndrome

Super-IgM syndrome results in impaired immunoglobulin class switching, leading to recurrent bacterial infections (*Streptococcus pneumoniae* and *Haemophilus influenzae*) and opportunistic infections like *Pneumocystis jirovecii* and *Cryptosporidium*. Prophylactic antibiotics and immunoglobulin replacement are critical. Newer therapeutic modalities, including gene editing techniques, aim to correct the genetic abnormalities underlying Super-IgM syndrome, offering hope for long-term disease resolution [87].

### 8.14. Chronic Mucocutaneous Candidiasis (CMC)

Chronic mucocutaneous candidiasis (CMC) is marked by recurrent infections of the skin, nails, and mucous membranes, primarily caused by *Candida albicans*. It results from immune defects, such as IL-17 signalling abnormalities or Th17 deficiencies, often linked to mutations like *STAT1* or *IL17RA*. CMC is associated with autoimmune conditions (e.g., APECED) or acquired immunodeficiencies like HIV/AIDS. Clinically, it presents as chronic oral thrush, vulvovaginal candidiasis, nail infections, and erythematous rashes. Severe cases may involve oesophageal candidiasis or secondary bacterial infections. While systemic infections are rare, they are life-threatening in immunocompromised individuals. Management involves antifungal therapy, including clotrimazole or fluconazole, and addressing underlying immune dysfunction with treatments such as interferon-gamma. Advances in understanding CMC have improved therapeutic strategies, significantly reducing disease burden [88].

### 8.15. Bruton’s Disease (X-Linked Agammaglobulinaemia, XLA)

Bruton’s disease, caused by *BTK* gene mutations, impairs B cell development, resulting in severely reduced immunoglobulin levels and humoral immunity. Patients are prone to recurrent respiratory and gastrointestinal infections from encapsulated bacteria such as *Streptococcus pneumoniae* and *Haemophilus influenzae*. Symptoms manifest early in life, often after maternal antibody protection wanes. Diagnosis involves detecting low immunoglobulin levels and absent B cells, confirmed by genetic testing. The treatment includes lifelong immunoglobulin replacement and prophylactic antibiotics, significantly reducing infections and improving quality of life [89].

### 8.16. Myasthenia Gravis (MG)

Myasthenia gravis (MG) is a chronic autoimmune disorder where autoantibodies target acetylcholine receptors, leading to fluctuating muscle weakness. Immunosuppressive therapies such as corticosteroids and azathioprine increase infection susceptibility. Common infections include pneumonia, which exacerbates muscle weakness, and opportunistic fungal infections. Poor swallowing mechanics can result in aspiration pneumonia. Management involves balancing immunosuppression, vaccination, and prophylactic antibiotics. Early diagnosis and infection control are crucial to reducing complications and improving outcomes [90].

### 8.17. Common Variable Immunodeficiency (CVID)

Common variable immunodeficiency (CVID) is a prevalent immunodeficiency marked by reduced immunoglobulin levels and impaired antibody production. Patients frequently suffer recurrent respiratory and gastrointestinal infections, particularly from *Streptococcus pneumoniae* and *Haemophilus influenzae*. Untreated infections may lead to bronchiectasis and chronic health complications. Chronic *Giardia lamblia* infections are also common, contributing to malabsorption. Management involves long-term immunoglobulin replacement therapy, which reduces infection frequency and improves quality of life [91].

### 8.18. Complement System Deficiencies

The complement system’s role in pathogen recognition and lysis makes its deficiencies significant. Classical pathway defects (e.g., C1q, C2, and C4) are linked to recurrent infections by encapsulated bacteria (*Streptococcus pneumoniae* and *Haemophilus influenzae*) and autoimmune disorders like systemic lupus erythematosus (SLE). Alternative pathway defects, including properdin deficiency, predispose individuals to *Neisseria meningitidis* infections. Terminal pathway deficiencies (C5–C9) impair membrane attack complex (MAC) formation, leading to recurrent Gram-negative bacterial infections. Management includes vaccination and prophylactic antibiotics, which significantly mitigate infection risks [92].

## 9. Study of the Microbiome in the Caribbean and Latin America

The microbiome, the collection of microorganisms residing in the human body, plays a pivotal role in health and disease. In the Caribbean and Latin America, the study of the microbiome has gained momentum, offering insights into its impact on regional public health challenges. These regions are characterised by unique cultural, dietary, and environmental factors that influence microbiome composition, making them a valuable area of research. Recent studies in the Caribbean have explored the gut microbiome’s role in non-communicable diseases such as diabetes and obesity, conditions with high prevalence in the region. Dietary habits, including probiotics and the traditional consumption of plant-based and seafood-rich diets, appear to modulate gut microbial diversity, potentially contributing to disease prevention. However, the increasing shift towards Westernised diets threatens this balance, with implications for health outcomes [93].

In Latin America, microbiome research has extended to infectious diseases, including dengue fever and Chagas disease. Investigations into the interplay between the microbiome and these diseases reveal the potential for microbiome-targeted interventions to improve treatment efficacy [94]. Moreover, studies on indigenous populations have provided invaluable data on microbiome diversity, which may serve as a benchmark for global comparisons [93]. Challenges persist, particularly limited funding, infrastructure, and expertise in genomics and bioinformatics [95]. Collaborative efforts between local researchers and international institutions have been crucial in overcoming these barriers. Initiatives such as the Latin American Microbiome Project underscore the importance of regional and global partnerships in advancing microbiome research [96].

The study of the microbiome in the Caribbean and Latin America is pivotal for understanding the complex interactions between human health, culture, and the environment. With sustained investment and collaboration, this research has the potential to revolutionise health interventions and improve disease management tailored to the unique needs of these regions. The microbiome in Latin America and the Caribbean represents a complex and diverse collection of microorganisms, including bacteria, viruses, fungi, and archaea, which inhabit the human body. These microbial communities are shaped by unique dietary patterns, environmental exposures, and cultural practices in the region. Traditional diets, often rich in plant-based foods and seafood, contribute to microbial diversity, which is linked to better health outcomes; however, the growing adoption of Westernised diets threatens this balance, leading to increased risks of non-communicable diseases such as diabetes and obesity [97].

Rising obesity and Type 2 diabetes pose global public health challenges, impacting quality of life and straining healthcare systems. Gut microbiota dysbiosis contributes to these conditions, influenced by diet, genetics, and treatments. This review examines the effects of pharmacotherapy and bariatric surgery on gut microbiota, highlighting microbiota-modulating strategies as promising non-surgical interventions for sustainable obesity management and prevention [98,99].

## 10. Short Discussion on Screening and Pathogen-Reduction Technologies Such as NAT and Other Techniques for the Detection of Viruses and Bacteria in Blood

Nucleic acid testing (NAT) has revolutionised blood transfusion safety by enabling the early detection of transfusion-transmissible infections (TTIs) at the molecular level [100]. Unlike traditional serological tests that detect antibodies or antigens, NAT directly identifies viral nucleic acids (RNA or DNA), significantly reducing the “window period”—the time between infection and the detectability of a pathogen [101]. This early detection capability is crucial for identifying blood-borne viruses such as HIV, hepatitis B virus (HBV), and hepatitis C virus (HCV), ensuring safer transfusions. The introduction of NAT has drastically reduced residual risks associated with TTIs, bringing transmission rates in some countries to less than one in a million for these pathogens [102]. Additionally, NAT enhances diagnostic sensitivity and specificity, minimising false negatives and ensuring infected blood is removed from the supply chain. It also allows for the detection of emerging and re-emerging pathogens, such as Zika virus and West Nile virus, bolstering the adaptability of blood safety systems to evolving threats [103,104].

NAT is particularly valuable in pooled-sample testing, where multiple donor samples are screened simultaneously, improving efficiency while maintaining high sensitivity. However, individual donor testing is preferred in high-risk settings or for specific pathogens such as HBV. Despite its advantages, NAT implementation presents challenges—especially in resource-limited settings—due to high costs, the need for advanced laboratory infrastructure, and the need for trained personnel [105]. Nevertheless, the long-term benefits of NAT, such as reduced healthcare costs from managing TTIs and increased public trust in transfusion safety, justify its adoption. Combining NAT with serological testing provides a comprehensive screening approach, addressing the limitations of each method and ensuring higher safety standards [106]. NAT has become an indispensable tool in modern transfusion medicine, improving safety, reducing risks, and enabling proactive responses to new pathogens [107,108]. Its continued expansion, even in resource-constrained environments, is essential for achieving safer blood transfusion systems globally.

In the table below are some advanced laboratory technologies for the detection of viruses and bacteria, along with their limitations (Table 1). While these technologies have advanced the field of microbial detection significantly, they also present challenges that must be addressed. Continuous innovation and research are necessary to improve their reliability, accessibility, and ease of use. Each method may be best suited for specific applications depending on the context of the infection being studied.

## 11. Global Infection Response

The global health system is composed of formal and informal networks of organisations across public, private-for-profit, and not-for-profit sectors, functioning at local, national, regional, and global levels to combat infectious disease threats. While this system has significantly advanced human health, it faces ongoing challenges from established, emerging, and re-emerging infectious diseases such as Ebola, Zika, dengue, MERS, SARS, influenza, and rising antimicrobial resistance. These challenges differ in severity, likelihood, and their impact on health and socio-economic outcomes, requiring varied responses from water sanitation initiatives to advanced biomedical interventions. Recent outbreaks, compounded by factors such as population growth in vulnerable areas, urbanisation, globalisation, climate change, civil unrest, and zoonotic pathogen transmission, have exposed critical gaps in the system’s effectiveness. Additionally, human-originated outbreaks from accidental laboratory releases or deliberate biological attacks present further risks. To address these multifaceted issues, the establishment of a multidisciplinary Global Technical Council on Infectious Disease Threats is recommended. This council would improve collaboration between key organisations, including the WHO, Gavi, CEPI, national disease control centres, and pharmaceutical companies. It would also address knowledge gaps by focusing on infectious disease surveillance, research needs, financing, supply chains, and the socio-economic impacts of outbreaks. By delivering high-level, evidence-based guidance, the council aims to enhance the global health system’s capacity to respond to infectious disease threats. This proposal envisions a more unified, proactive, and effective framework for managing the diverse risks posed by infectious diseases, ultimately reducing their health and socio-economic impacts worldwide [123].

## 12. Development of Three Experimental Vaccine Candidates Targeting HIV, *Salmonella*, and *Staphylococcus aureus*

### 12.1. Understanding Anti-Idiotypic Antibodies

Anti-idiotypic antibodies are a unique subset of antibodies that bind to the antigen-binding sites of other antibodies (idiotypes) [124,125,126]. In this context, the following applies:Idiotype: the unique antigen-binding region of an antibody;Anti-idiotypic antibodies: these are secondary antibodies that recognise and bind to the idiotype of the initial antibody.

Anti-idiotypic antibodies can mimic the structure of the original antigen. This mimicry allows them to stimulate an immune response similar to that induced by the actual antigen, serving as a “surrogate antigen”. This property forms the basis of their potential use as vaccines.

### 12.2. Production of Anti-Anti-Idiotypic Antibodies in Chickens

In this study, the production of anti-anti-idiotypic antibodies occurred as follows:Chickens were immunised with an antigen of interest;This led to the production of specific antibodies (the primary response) against the antigen;Anti-idiotypic antibodies were then developed against these primary antibodies, mimicking the original antigen’s structure;Subsequently, anti-anti-idiotypic antibodies (tertiary response) were produced in the chickens. These tertiary antibodies recognise the anti-idiotypic antibodies, essentially recapitulating the original antigenic structure.

### 12.3. Role as Vaccines

Anti-anti-idiotypic antibodies act as a self-sustaining immunogenic system:These antibodies effectively mimic the native antigen without requiring the actual pathogen or antigen in vaccine formulations;They stimulate both humoral (antibody-mediated) and cellular immune responses;This strategy is particularly valuable when the original antigen is difficult to synthesise, unstable, or associated with high production costs.

### 12.4. Applications and Benefits

Pathogen mimicry: they provide a safe method of immunisation by avoiding live or inactivated pathogens;Cross-species applicability: as chickens produce a robust immune response, this model can guide vaccine development for other species, including humans;Cost-effectiveness: they eliminate the need for pathogen culture or synthetic antigen production.

This study demonstrates the feasibility of leveraging anti-idiotypic and anti-anti-idiotypic antibodies as innovative, effective, and scalable vaccine candidates.

### 12.5. Specific Details

The experiments described in this study may represent the first report of a chicken-and-egg system for the production of anti-HIV-gp120 antibodies. Eggs from chickens immunised with a specific immunogen might be considered as a special type of oral anti-idiotypic vaccine. This was shown by the development of anti-anti-HIV gp120 antibodies in cats fed anti-HIVgp120-positive eggs. The presence of these antibodies could prove useful in exploring the development of an oral active immunisation to fight HIV in asymptomatic carriers, chronic HIV patients, and in general, a vast amount of infectious diseases. The authors are not aware of any other documented studies describing the use of anti-HIV-positive eggs as an oral anti-idiotypic vaccine.

A phase II trial tested the idiotype vaccine mAb 13B8.2 in early-stage HIV+ patients with 350–500 CD4+ cells/μL. The monoclonal antibody targets the CD4/D1 region involved in HIV-gp120 binding and previously demonstrated cross-reactive immunity in vitro. A total of 158 patients were randomised to receive either 1.2 mg of alum-precipitated mAb 13B8.2 or placebo. The vaccine was well-tolerated, with mainly local side effects. The treated group showed significantly higher HIV-1 neutralisation titres and gp120 binding titres, suggesting enhanced anti-viral immune responses and potential benefits for HIV disease management [127]. This study examined whether repeated oral mucosal stimulation with HIV-Immunosomes induces secretory IgA in saliva and primes the immune system for a rapid systemic response to subsequent parenteral immunisation with low antigen doses. In rabbits, HIV-1 gp160-specific secretory IgA was detected in saliva following oral immunisation. After parenteral immunisation, mice and rabbits pre-exposed orally exhibited high serum IgA, IgM, and IgG titres within a week, neutralising HIV infectivity in vitro. In contrast, parenteral-only immunisation induced a significantly weaker humoral immune response [128]. These antibodies neutralised HIV infectivity in vitro.

### 12.6. Immunogenicity Studies of Experimental HIV Vaccine in Chickens [129]

All reagents used in this study were commercially available and purchased from Sigma-Aldrich (St. Louis, MO, USA). Each experiment was performed in triplicate, yielding consistent results. The HIV immunogens were keyhole limpet haemocyanin (KLH) conjugated to HIV peptides, specifically fragments 308–331 and 421–438 of HIV gp120 and fragment 579–601 of HIV gp41. The amino acid sequences of these peptides were as follows:HIV-gp41 (579–601): Arg-Ile-Leu-Ala-Val-Glu-Arg-Tyr-Leu-Lys-Asp-Gln-Gln-Leu-Leu-Gly-Ile-Trp-Gly-Cys-Ser-Gly-Lys [130];HIV-gp120 (308–331): Asn-Asn-Thr-Arg-Lys-Ser-Ile-Arg-Ile-Gln-Arg-Gly-Pro-Gly-Arg-Ala-Phe-Val-Thr-Ile-Gly-Lys-Ile-Gly [131];HIV-gp120 (421–438): Lys-Gln-Phe-Ile-Asn-Met-Trp-Gln-Glu-Val-Gly-Lys-Ala-Met-Tyr-Ala-Pro-Pro [132].

### 12.7. Dimerisation of HIV Peptides and Immunogen Preparation

The HIV peptides were modified by adding a C-terminal cysteine for dimerisation via cysteine oxidation using dimethyl sulfoxide. Each peptide was dissolved in 5% acetic acid at 5.1 mg/mL, adjusted to pH 6 with ammonium carbonate, and treated with dimethyl sulfoxide at 20% of the final volume for 4 h at room temperature. The solutes were precipitated with trifluoroacetic acid and cold ether, dialysed against deionised water (pH 7) at 4 °C overnight, and stored at 4 °C [129].

KLH was diluted in borate buffer (pH 10) and conjugated with 1.1 μmol of each HIV peptide using 0.3% glutaraldehyde. The reaction was blocked with glycine, and the conjugates were dialysed against borate buffer (pH 8.4) overnight [129].

### 12.8. Chicken Immunisation

Six healthy brown Leghorn hens (two per immunogen) were immunised intramuscularly with 0.5 mg/mL of the KLH-conjugated HIV peptide in Freund’s complete adjuvant on day 0, followed by booster doses of 0.25 mg/mL in Freund’s incomplete adjuvant on days 14, 28, and 45. Eggs were collected daily pre- and post-immunisation. Additional hens (*n* = 15) were similarly immunised to evaluate ELISA reproducibility. The water-soluble fraction (WSF) containing IgY was isolated using a modified Polson method [129].

### 12.9. ELISA for Anti-HIV Antibodies

Microplates were coated with HIV peptides (100 ng/well) in coating buffer, washed, and blocked with non-fat milk in PBS. WSF samples diluted to 1:50 were added, followed by horseradish peroxidase-conjugated anti-IgY (diluted to 1:30,000). After incubation with TMB substrate, absorbance was measured at 450 nm. Cut-off values were 0.42, 0.40, and 0.44 for peptides 579–601, 308–331, and 421–438, respectively [129]. Positive and negative controls were prepared from immunised and non-immunised hens, respectively [129].

### 12.10. Immunogenicity Studies of Salmonellosis Experimental Vaccine in Chickens: Preparation and Immunisation

The immunogen comprised live-attenuated salmonella serovars: *Montevideo*, *Yeerongpilly*, *Augustenborg*, *Kentucky*, and *Typhimurium.* Birds (*n* = 30 per group) were immunised intramuscularly or orally with the immunogen and booster doses on days 14 and 28. Controls received normal saline solution with adjuvant. The WSF from eggs collected pre- and post-immunisation was assayed for anti-*Salmonella* antibodies using ELISA [129].

### 12.11. Experimental Infection and Analysis [129]

Forty-five days post-immunisation, birds were challenged with a wild-type multidrug-resistant *Salmonella Typhimurium* strain (103 CFU/g). Immunised birds exhibited reduced infection rates and increased specific antibody production. Statistical analysis confirmed significant differences (*p* < 0.05) between the groups.

### 12.12. Antibiotic Susceptibility Testing

Salmonella isolates were tested using the disc diffusion method against various antibiotics, including gentamicin, ciprofloxacin, and tetracycline. Multidrug resistance was confirmed.

### 12.13. Immunogenicity Studies of Experimental Staphylococcus aureus Vaccine [129]

Hens were immunised with Staphylococcal protein A (SpA) using Freund’s adjuvant. ELISA confirmed anti-SpA antibody (Ab-1) production in WSF. Chicks fed hyperimmune eggs demonstrated Ab-3-mediated inhibition of *S. aureus* growth. Competitive ELISA and affinity chromatography confirmed functional Ab-2 and Ab-3 interactions. Statistical analyses showed significant bacterial growth inhibition in hyperimmune groups (*p* < 0.05).

This study demonstrated the feasibility of inducing specific antibody responses against HIV, *Salmonella*, and *S. aureus* in chickens, supporting the potential for vaccine development and passive immunisation strategies.

### 12.14. Results and Discussion

Table 2 displays the results of the water-soluble fractions (WSF) of egg yolks collected at day 0 and day 60 post-immunisation. These were assayed for the presence of anti-HIV peptide antibodies using ELISA. Each sample (six in total) was tested three times. The results demonstrated that HIV peptide vaccines effectively induced strong anti-HIV immune responses in immunised brown Leghorn layer hens. A statistically significant difference (*p* < 0.01) was observed in the anti-HIV antibody levels between pre- and post-immunisation birds across all three experimental vaccines [129].

These results were obtained from six healthy brown Leghorn hens (two per vaccine) immunised intramuscularly (IM) at multiple breast sites. Eggs were collected pre- and post-immunisation. The WSF of each egg yolk was isolated using the Polson method (1990) and tested via ELISA. The intra- and inter-assay coefficients of variation were within acceptable limits, indicating assay reproducibility.

Table 3 summarises the ELISA results for anti-*Salmonella* antibodies in brown Leghorn layer hens immunised via intramuscular (IMI) and oral (OI) routes. A placebo group was included as chickens may naturally possess anti-*Salmonella* antibodies due to environmental exposure. Results indicated high titres of specific antibodies in immunised birds compared to controls, with statistically significant differences (*p* < 0.01).

Eggs were collected pre- and post-immunisation, and egg yolk proteins were assessed for anti-*Salmonella* antibodies. The intra- and inter-assay coefficients of variation were less than 5% and 10%, respectively.

Table 4 reports the results of a challenge study with wild-type *Salmonella Typhimurium*. Immunised groups (IMI and OI) showed significantly lower *Salmonella* colonisation in the caeca and stomach compared to controls (*p* < 0.01), indicating the effectiveness of the vaccine.

These findings highlight that vaccinated birds developed protective antibodies, effectively combating *Salmonella* infection, whereas the control group showed higher susceptibility due to the absence of specific antibodies.

The results of this study demonstrate the strong immunogenicity and protective efficacy of the experimental *Salmonella* vaccine in brown Leghorn layer hens. ELISA analysis revealed significantly higher titres of anti-*Salmonella* antibodies in both intramuscularly (IMI) and orally immunised (OI) groups compared to the placebo control group, with statistically significant differences observed post-immunisation (*p* < 0.01). This highlights the robust antibody response induced by the vaccine, irrespective of the administration route. Notably, the placebo group exhibited minimal antibody titres, likely due to natural environmental exposure to *Salmonella*, although these levels were not sufficient for significant protection [129].

Egg yolk antibody assessment further validated the vaccine’s immunogenicity, with intra- and inter-assay variation coefficients maintained below 5% and 10%, ensuring reliability. Importantly, the challenge study confirmed the vaccine’s protective efficacy, as birds in the immunised groups (IMI and OI) displayed substantially lower *Salmonella* colonisation in both the caeca and stomach compared to the control group (*p* < 0.01). Specifically, bacterial concentrations in vaccinated birds remained at or below 10^2^ CFU/g, while the control group exhibited high levels of colonisation, with concentrations reaching up to 2 × 10^7^ CFU/g in the caeca [129].

Further research should explore the long-term durability of the immune response, optimal dosing strategies, and potential scalability of this vaccine for commercial poultry production. These results provide a promising foundation for enhancing *Salmonella* control measures in poultry, contributing to food safety and public health improvements. The results demonstrate that polyclonal antibodies produced in immunised chickens exhibit potent inhibitory effects against HIV, *Salmonella*, and *Staphylococcus aureus*. Specifically, the anti-HIV antibody responses were strong and reproducible, as evidenced by significantly higher antibody titres in the post-immunisation samples. Similarly, the *Salmonella* vaccine induced robust immune responses, which effectively reduced bacterial colonisation in critical organs, such as the caeca and stomach, following a challenge with a wild-type strain. The inhibition of *S. aureus* growth by anti-idiotypic antibodies (Ab-3) highlights the potential of hyperimmune eggs for providing protective immunity. Figure 3, Figure 4 and Figure 5 illustrate the efficacy of Ab-3 in neutralising *S. aureus* colony formation across various time points, demonstrating the longevity and functional activity of these antibodies. These results suggest that Ab-3, acting as a mirror image of SpA, binds to the bacterial cell wall and inhibits its growth. The absence of such inhibition in non-fed and control groups underscores the specificity and efficacy of the immunised-derived antibodies. The findings from this study also align with Jerne’s network theory [129], where successive generations of idiotypic and anti-idiotypic antibodies interact to produce a robust immune response. This mechanism underscores the potential for using avian antibodies in developing innovative immunotherapies for infectious diseases. Furthermore, the passive transfer of immunity via hyperimmune eggs could provide a non-invasive and sustainable approach for managing bacterial and viral infections in both animals and humans. Future research should focus on further characterisation of these antibodies, including their epitope specificity, binding kinetics, and long-term stability. Additionally, the exploration of oral immunisation strategies, as demonstrated with the *Salmonella* vaccine, presents a promising avenue for scalable and cost-effective vaccine delivery systems. Egg-yolk-derived antibodies (IgY) offer several advantages over mammalian antibodies, including enhanced animal welfare, high productivity, and improved specificity. Their lack of reactivity with the human complement system and B-cell repertoire makes IgY particularly suitable for therapeutic and diagnostic applications. The ability to non-invasively harvest large quantities of antibodies further enhances their utility. These attributes position IgY as a valuable tool in the fight against infectious diseases, with potential applications ranging from diagnostics to therapeutic interventions [133,134].

Egg yolk serves as a critical and alternative source of polyclonal antibodies, offering several advantages over mammalian antibodies in terms of animal welfare, productivity, and specificity. The primary antibody in avian blood, immunoglobulin Y (IgY), is transmitted to their offspring and stored in the egg yolk, enabling the non-invasive collection of large quantities of antibodies. Additionally, due to phylogenetic differences and unique structural molecular features, IgY is particularly suitable for diagnostic and therapeutic applications. This suitability stems from its lack of reactivity with the human complement system and B-cell repertoire. However, IgY exhibits higher avidity for conserved mammalian polypeptides. Chicken-derived IgY antibodies have demonstrated broad applicability in both therapeutic and diagnostic settings. They are used in the diagnosis of viral, bacterial, and fungal infections, as well as in the treatment of infectious diseases in humans and animals [129].

Figure 5 illustrates the inhibition of *Staphylococcus aureus* growth by Ab-3, which was purified from pooled sera of six chicks fed with anti-SpA hyperimmune eggs over an 8-day period. It was observed that *S. aureus* growth inhibition occurs in the treated dilutions, indicating that Ab-3 acts as a mirror image of SpA. By binding to SpA on the cell wall of *S. aureus*, Ab-3 effectively inhibits the pathogen’s growth. Notably, this effect was not seen in non-fed chicks or control groups, highlighting the importance of the hyperimmune diet in enhancing the chicks’ immunological response [129].

The results are depicted as follows:The *Y*-axis, representing bacterial concentration (mL), shows absorbance values ranging from 0.2 to 1.4, indicating varying levels of inhibition at decreased concentrations of bacteria (0.1, 0.01, and 0.001 on the *X*-axis);Absorbance was carefully measured at 600 nm, providing quantifiable evidence of Ab-3’s effect on bacterial growth.

Through several experiments, it was confirmed that both Ab-1 and Ab-2 coexisted within the egg yolks. These antibodies were effectively separated from the water-soluble fraction (WSF) by utilising affinity chromatography techniques. A total of six preparations were made, and the reproducibility of the indirect ELISA assays for the determination of anti-anti-idiotypic salmonella antibodies (Ab-3) indicated high reliability, as evidenced by consistent coefficient of variation (CV) values across trials. Significant insights were gained regarding the utility of hyperimmune avian egg yolk. Chicks that ingested eggs from chickens immunised with anti-SpA antibodies exhibited strong and specific immune responses to *S. aureus*. This suggests that the immunoglobulins obtained from hyperimmune eggs may serve as a promising alternative therapeutic strategy for enhancing resistance against bacterial pathogens. Looking forward, future research will aim to purify and characterise anti-salmonella and anti-HIV antibodies by employing similar methodologies to those successfully used for anti-SpA antibodies. The investigation will also explore the mechanisms of immunity conferred by these antibodies, further elucidating their potential roles in protective immunity. This discovery opens up new perspectives for immunotherapy targeting both existing and emerging infectious diseases. The preliminary results provide promising avenues for the development of oral vaccines against common and neglected pathogens, thereby addressing gaps in current vaccine strategies. Further exploration of hyperimmune egg yolk may expand its role in veterinary and human medicine, ultimately contributing to improved health outcomes through innovative vaccination approaches. Anti-idiotypic antibodies (anti-Id Abs) are a promising therapeutic approach for cancer and autoimmune diseases due to their ability to mimic antigens or regulate immune responses. In autoimmune diseases, anti-Id Abs specifically neutralise pathogenic autoantibodies and restore immune tolerance without broadly suppressing the immune system. This targeted approach shows promise in conditions like systemic lupus erythematosus and rheumatoid arthritis by reducing inflammation and tissue damage. Despite challenges such as production complexity, safety concerns, and the need for personalised strategies, advancements in biotechnology and immunology are driving their development. Anti-idiotypic antibodies represent a significant step towards precision medicine, offering innovative and highly specific treatments for immune-mediated diseases [133,134,135,136,137,138,139,140,141].

## 13. Conclusions

This comprehensive study delves into the critical challenges of infectious diseases in the Caribbean and South America, particularly tuberculosis and nosocomial infections, and their intersections with immunodeficiency, antibiotic resistance, and the microbiome. The high HIV-TB co-infection rates in Jamaica and Trinidad and Tobago underscore the need for early detection and integrated treatment strategies. Furthermore, the spread of antibiotic-resistant pathogens, such as Gram-negative bacteria and MRSA, poses a significant threat in healthcare settings, necessitating stringent infection control measures and prudent antibiotic use. The findings related to *Salmonella* infections in poultry in Jamaica highlight the importance of biosecurity measures and public education to reduce foodborne disease risks.

Additionally, this study emphasises the vulnerability of immunodeficient patients, such as those with SCID, CGD, and other immunological disorders, to various infections. In conclusion, a multidisciplinary approach involving molecular characterisation, immunological research, and effective public health strategies is essential for controlling infectious diseases in the region, particularly in resource-limited settings. Primary immunodeficiency disorders (PIDs) and autoimmune conditions in South America and the Caribbean exhibit diverse immunological characteristics due to genetic, environmental, and infectious factors. These immunological disorders pose significant health challenges in South America and the Caribbean, highlighting the need for tailored diagnostic and therapeutic approaches.

Antimicrobial resistance (AMR) remains a critical public health challenge in the Caribbean and Latin America, exacerbated by improper antibiotic use, weak regulatory frameworks, and limited public awareness. This review underscores the importance of strengthening surveillance systems, implementing antibiotic stewardship programmes, and enhancing healthcare access to mitigate AMR’s impact.

### 13.1. Actionable Insights

To achieve these goals, policymakers must prioritise the following:Establishing regionally coordinated surveillance networks like ReLAVRA to monitor resistance trends;Enforcing stricter regulations to limit over-the-counter sales of antibiotics and combat counterfeit medications;Promoting public awareness campaigns tailored to cultural and regional contexts to educate communities about the responsible use of antibiotics.

Healthcare providers should integrate rapid diagnostic testing to guide appropriate antibiotic prescriptions, while investments in healthcare infrastructure and research into alternative therapies must be expanded.

### 13.2. Future Research Directions

Investigating the socio-economic and behavioural drivers of antibiotic misuse in urban and rural settings;Exploring novel treatments, including immunomodulators, and anti-idiotypic vaccines to address infections resistant to conventional antibiotics;Assessing the environmental impact of antibiotics in agricultural and industrial runoff, identifying measures to mitigate contamination.

A collaborative approach, leveraging regional partnerships and international support, is essential to overcoming the growing AMR crisis. By implementing these strategies, the region can protect its populations from the devastating effects of AMR while contributing efforts to curb this threat.

The microbiome plays a critical role in Latin America and the Caribbean, influencing health, disease prevention, and environmental resilience, offering opportunities for advancing regional public health and sustainability strategies.

## Figures and Tables

**Figure 1 microorganisms-13-00282-f001:**
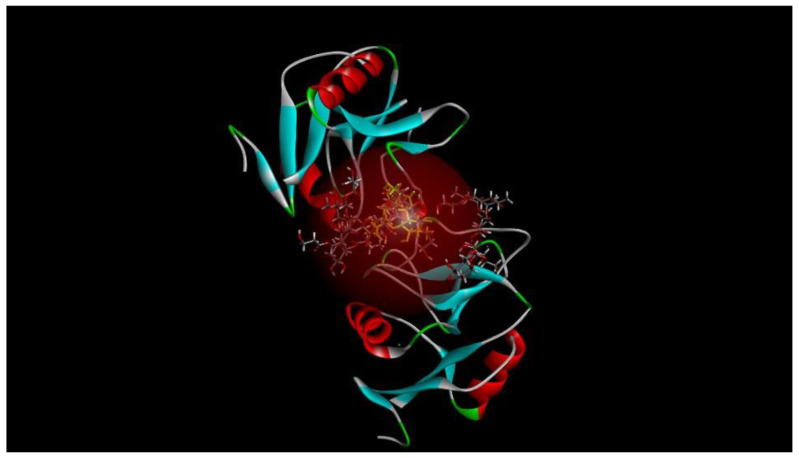
The diagram above shows the five most feasible poses in which trehalose dibenzenate, the derivative analogue of trehalose dimycolate, binds to mincle because of the low force field energy shown above in a reddish hemisphere, which is the area in which most hydrogen bond interactions occur [67].

**Figure 2 microorganisms-13-00282-f002:**
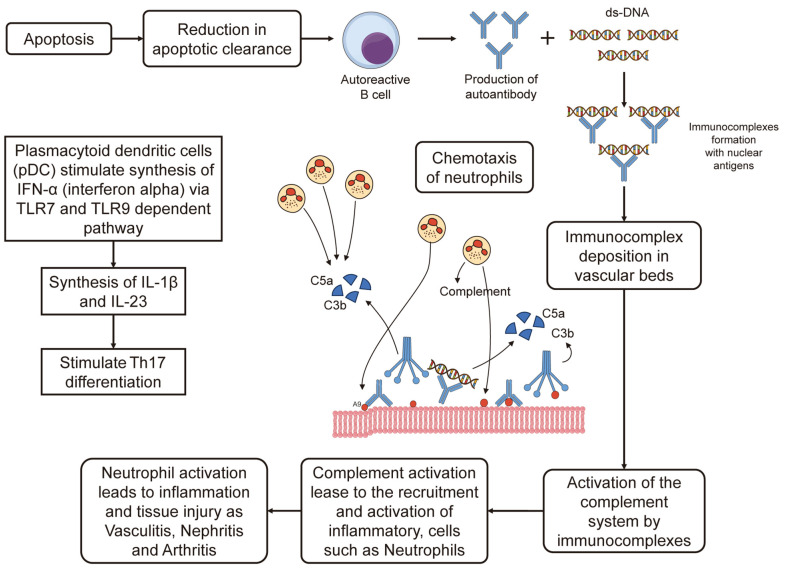
In systemic lupus erythematosus (SLE), a type III hypersensitivity reaction plays a crucial role. This process involves the formation of immune complexes, which subsequently activate the complement system. Complement components such as C3a and C5a act as potent chemotactic factors, attracting neutrophils to the sites where immune complexes are deposited. The activation of these neutrophils leads to local inflammation and tissue damage, contributing to conditions like vasculitis, nephritis, and arthritis. In addition to complement activation, cytokines such as TNF-α, IL-6, and IFN-γ are also involved, further amplifying the inflammatory response and perpetuating tissue injury. Other mechanisms may also contribute to the pathology. Taken from [73]. (EB16 For English services to fix what is in the box as the English editor suggests).

**Figure 3 microorganisms-13-00282-f003:**
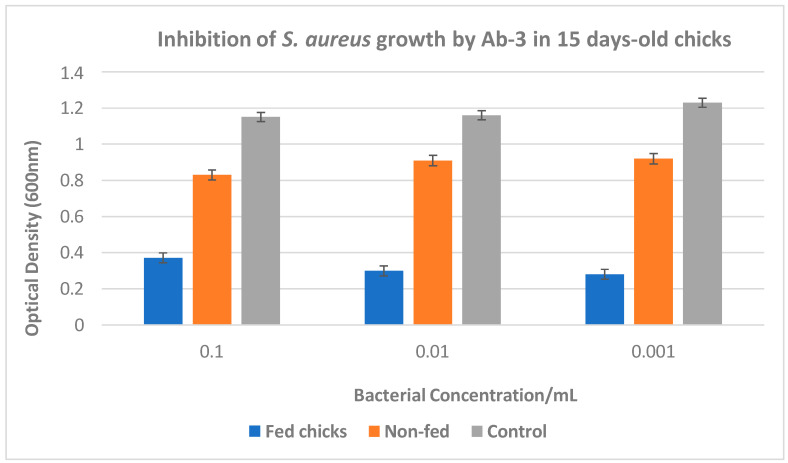
Inhibition of *S. aureus* growth by Ab-3 purified by SpA-affinity chromatography and observed in pooled sera of chicks (six) fed with anti-SpA hyperimmune eggs up to 15 days old. Inhibition was observed in dilutions of *S. aureus* treatment, suggesting that the Ab-3 is a mirror image of SpA, and binding to it on the cell wall of *S. aureus* inhibits the pathogen growth; this does not happen in non-fed chicks and controls [129]. Control: bacterial suspension without Ab-3. Non-fed animal: fed with non-hyperimmune eggs (eluates from purification, which tested negative for the presence of Ab-3 by the sandwich ELISA). The graph depicts the relationship between optical density at 600 nm (OD600) and bacterial concentration per mL, comparing the inhibition of *S. aureus* growth among fed chicks, non-fed chicks, and the control group.

**Figure 4 microorganisms-13-00282-f004:**
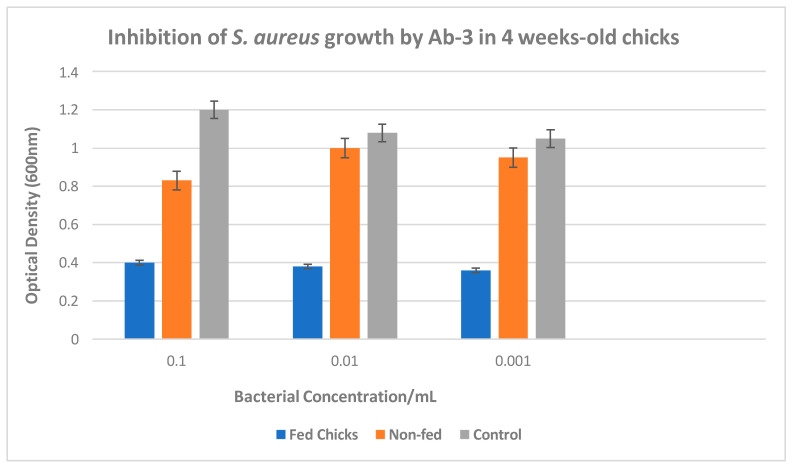
Inhibition of *S. aureus* growth by Ab-3 purified by SpA-affinity chromatography and observed in pooled sera of chicks (six) fed with anti-SpA hyperimmune eggs up to 4 weeks. Inhibition was observed in dilutions of *S. aureus* treatment, suggesting that the Ab-3 is a mirror image of SpA, binding to it on the cell wall of *S. aureus* inhibits the pathogen growth; this does not happen in non-fed chicks and controls [129].

**Figure 5 microorganisms-13-00282-f005:**
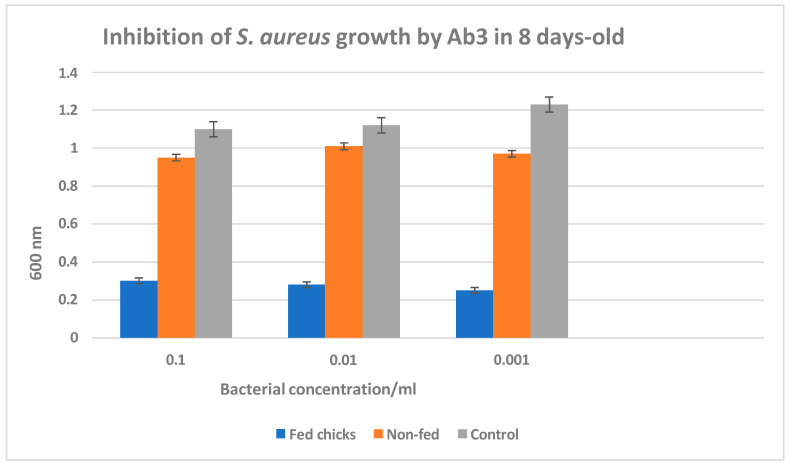
Inhibition of *S. aureus* growth by Ab-3 purified by SpA-affinity chromatography and observed in pooled sera of chicks (six) fed with anti-SpA hyperimmune eggs up to 8 days. Inhibition was observed in dilutions of *S. aureus* treatment, suggesting that the Ab-3 is a mirror image of SpA, binding to it on the cell wall of *S. aureus* inhibits the pathogen growth; this does not happen in non-fed chicks and controls [129]. EB34 add optical density.

**Table 1 microorganisms-13-00282-t001:** Nucleic acid technology (NAT) and other techniques for detection of viruses and bacteria in blood.

Advanced Laboratory Techniques	Description	Limitations	References
Polymerase Chain Reaction (PCR)	PCR amplifies specific DNA sequences, making it easier to detect even small amounts of genetic material.	Requires prior knowledge of the virus/bacteria’s genetic sequence. Contamination can lead to false positives. Time-consuming and requires skilled personnel. Equipment costs can be high.	[109]
Next-Generation Sequencing (NGS)	NGS can sequence entire genomes, allowing for the identification of viruses and bacteria at a genomic level.	High cost and complex data analysis requirements. Requires extensive bioinformatics support to interpret results. Not all pathogens are covered in existing databases, which may lead to unidentified organisms.	[110,111]
Enzyme-Linked Immunosorbent Assay (ELISA)	ELISA detects specific proteins (antigens) associated with pathogens using enzyme-linked antibodies.	May not distinguish between current and past infections (i.e., presence of antibodies). Cross-reactivity can occur, leading to false positives. Requires specific antibodies, which may not always be available.	[112,113]
Mass Spectrometry (MS)	MS identifies and quantifies biomolecules, including proteins and nucleic acids, based on their mass and charge.	Complex sample preparation required. Can be less effective for detecting low-abundance pathogens. Interpretation of results can be challenging and requires expert knowledge.	[114,115]
Lateral Flow Immunoassay (LFIA)	A rapid test that uses antibodies to detect the presence of specific pathogens in a sample.	Generally lower sensitivity and specificity compared to more advanced methods. Results can be qualitative rather than quantitative. Subject to user error and environmental factors can affect results.	[116,117]
Digital PCR (dPCR)	An evolution of traditional PCR that partitions the sample into many small reactions to provide more precise quantification.	Requires specialised equipment and can be costly. Limited throughput compared to standard PCR methods. Interpretation and data analysis can be complex.	[118,119]
CRISPR-Based Detection Technologies	Utilise CRISPR technology for specific detection of nucleic acids with high sensitivity and specificity.	Still a developing technology with limited commercial availability. Potential for unexpected off-target effects, leading to inaccuracies. Requires technical expertise for setup and interpretation.	[120,121,122]

**Table 2 microorganisms-13-00282-t002:** Results of immunogenicity studies of experimental HIV vaccine in brown Leghorn layer hens [129].

ELISA Results of the Experimental Vaccines	Mean Optical Density (XOD) ± SD Pre-Immunisation (Day 0)	Mean Optical Density (XOD) ± SD Post-Immunisation (Day 60)	*p*-Value
HIV-gp41 (579–601)	0.170 ± 0.021	0.885 ± 0.044	*p* < 0.01
HIV-gp120 (308–331)	0.156 ± 0.015	0.910 ± 0.024	*p* < 0.01
HIV-gp41 (421–438)	0.188 ± 0.010	0.865 ± 0.038	*p* < 0.01

**Table 3 microorganisms-13-00282-t003:** Results of immunogenicity studies of experimental *Salmonella* vaccine in brown Leghorn layer hens [129].

Route of Administration	Mean Optical Density (XOD) ± SD Pre-Immunisation	Mean Optical Density (XOD) ± SD Post-Immunisation (Days 40–42)	*p*-Value
Intramuscularly Immunised (IMI, N = 30)	0.310 ± 0.059	1.481 ± 0.139	*p* < 0.01
Orally Immunised (OI, N = 30)	0.359 ± 0.110	1.585 ± 0.156	*p* < 0.01
Placebo Control Group (CG, N = 30)	0.303 ± 0.067	0.365 ± 0.053	*p* < 0.05

**Table 4 microorganisms-13-00282-t004:** *Salmonella* contamination percentage in caeca and stomach after challenge with wild-type *Salmonella Typhimurium* [129].

Group	Mean Concentration of Salmonella in Caeca (CFU/g)	Mean Concentration of Salmonella in Stomach (CFU/g)	*p*-Value
Live-Attenuated Vaccine (N = 30)	10^2^	10^2^	*p* < 0.01
Control Group (N = 30)	2 × 10^7^	10^5^	*p* < 0.01

## Data Availability

The dataset supporting the findings of this study is included within the manuscript and its referenced sources, ensuring comprehensive access to the relevant data for further examination and analysis.

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
