# Peer review of "Tackling Infectious Diseases in the Caribbean and South America: Epidemiological Insights, Antibiotic Resistance, Associated Infectious Diseases in Immunological Disorders, Global Infection Response, and Experimental Anti-Idiotypic Vaccine Candidates Against Microorganisms of Public Health Importance"

_microorganisms, 2025, doi:10.3390/microorganisms13020282_

Round 1

Reviewer 1 Report

Comments and Suggestions for Authors

Dear Editor,

Thank you for the opportunity to review the manuscript titled "Tackling Infectious Diseases in the Caribbean and South America: Epidemiological Insights, Antibiotic Resistance, and Associated Immunological Diseases." Below are my comments and recommendations for improvement.

Streamline the manuscript by prioritizing topics like TB and HIV co-infection, antibiotic resistance, and immunodeficiency-related infections. Provide a more focused and in-depth analysis of these areas.

Improve the clarity and organization of figures and tables.

Author Response

Thank you for your valuable feedback on the manuscript titled "Tackling Infectious Diseases in the Caribbean and South America: Epidemiological Insights, Antibiotic Resistance, and Associated Immunological Diseases." I appreciate your insights and would like to let you know that all of the issues you raised have been addressed.

We have streamlined the content to prioritize topics such as TB and HIV co-infection, antibiotic resistance, and immunodeficiency-related infections, providing a more focused and in-depth analysis of these critical areas. Additionally, we have enhanced the clarity and organization of figures and tables to improve overall readability and comprehension.

Thank you again for your thoughtful input, which has helped strengthen the manuscript.

Reviewer 2 Report

Comments and Suggestions for Authors

Dear authors,

As a result of reading your manuscript, I have a number of questions:

1.    The purpose of the study is missing.

2.    The introduction needs to be expanded. The problem is not clearly described, it is not clear from the introduction why such a review was necessary. The novelty is also not stated.

3.    Section “Nosocomial Infections Globally” – It is necessary to add statistics for at least the last 3-5 years. Studies from 1992-1995 and 2005 are very old data. And there is too little information on global studies for this subsection.

4.    Lines 68-70 – “In Trinidad and Tobago, limited data exist, but between 1992-1995, 7,158 infections were recorded, primarily urinary tract infections, postoperative wound infections, pneumonia, and bloodstream infections.” – You need to add a link to the publication where this stat is published. 

5.    Lines 52-53 – “Peptide candidate vaccines to chicks induces specific anti-HIV gp120 and gp41 antibodies…” – is that line in italics on purpose?

6.    Section “3. Salmonellosis” – the characterisation of Salmonella bacteria should be added.

7.    Lines 141-145 – “Among these, the risk of transfusion-transmitted bacterial infections (TTBI) is notably higher than that of viral infections, particularly in platelet transfusions, which present the highest risk of bacterial contamination. This makes blood safety a major concern in healthcare systems worldwide [32-37].” – you refer, among other things, to the work of [35]  - Justiz-Vaillant, A.A.; Akpaka, P.E.; McFarlane-Anderson, N.; Smikle, M.P. Comparison of Techniques of Detecting Immunoglobulin-Binding Protein Reactivity to Immunoglobulin Produced by Different Avian and Mammalian Species. West Indian Med. J. 2013, 62, 12–20. However, the content of this reference does not match what you write in lines 141-145. The reference is incorrect.

8.    Line 145 - Reference number [36]  Justiz-Vaillant, A.A. A Protocol and Detailed Methodological Study on Immunogenicity of Various Experimental Vaccines. Int. 559 Biol. Biomed. J. 2021, 7, 1–10. Available online: http://ibbj.org/article-1-252-fa.html (accessed on Sept 11, 2024]) – is also not correct.

9.    Line 159 – Reference number [39] - Justiz-Vaillant, A.A. Multiple Myeloma Update. Int. Biol. Biomed. J. 2018, 4, 136–141. Available online: http://ibbj.org/article-1-177-en.html (accessed on Sept 12, 2024]). –  Reference is incorrect, does not match the information in lines 154-159.

10.                       Line 168 – Reference number [46] - Vaillant, A.J.; Bazuaye, P.; McFarlane-Anderson, N.; Smikle, M.P.; Fletcher, H.; Akpaka, P.E. Association Between ABO Blood  Type and Cervical Dysplasia/Carcinoma in Jamaican Women. Br. J. Med. Med. Res. 2013, 3, 2017–2021. – Reference is not relevant to the information in Lines 160-168. 

11.                       Lines 164-165 – “There is association between some cancers and ABO groups.” – This information is not relevant to the issue of blood-borne infections under discussion.

12.                       Section “5. Egg Antibody Technology” – Information should be included to provide a historical background to the development of the idea of studying egg antibody technology. It is also necessary to justify the presence of this section in the manuscript. 

13.                        Section “6. Confronting Tuberculosis”- References to very old publications.

14.                       Lines 233-234 “The average age in both countries was around 38 years, and male-to-female ratios were 2.25 in Guyana and 4.27 in Suriname” – What is this information for?

15.                       Line 252 – Reference [64] - Soodeen, S.; Justiz-Vaillant, A.; Jalsa, N. Is Possible Molecular Docking of Carbohydrates to a Mycobacterium Tuberculosis Molecule? Preprints 2023. Available online: https://www.preprints.org (accessed on [date of access]). – It is not correct to refer to a preprint in a review.

16.                       Subsection “7.1. Severe Combined Immunodeficiency Disorders (SCID)” – you only used one reference to your review - [66] - Justiz-Vaillant, A.A.; Gopaul, D.; Akpaka, P.E.; Soodeen, S.; Arozarena-Fundora, R. Severe Combined Immunodeficiency— Classification, Microbiology Association and Treatment. Microorganisms. 2023, 11, 1589. https://doi.org/10.3390/microorganisms11061589.  This is not correct.

17.                       Subsection “7.2. Chronic Granulomatous Disease (CGD)” – you only used one reference to your review [67] Justiz-Vaillant, A.A.; Williams-Persad, A.F.-A.; Arozarena-Fundora, R.; Gopaul, D.; Soodeen, S.; Asin-Milan, O.; Thompson, R.; Unakal, C.; Akpaka, P.E. Chronic Granulomatous Disease (CGD): Commonly Associated Pathogens, Diagnosis and Treatment. Microorganisms. 2023, 11, 2233. https://doi.org/10.3390/microorganisms11092233. This is not correct.

18.                       Subsection “7.3. Transient Hypogammaglobulinemia of Infancy (THI)” – You've only used one source, and that's the reference of your review. [68]  Justiz-Vaillant, A.A.; Hoyte, T.; Davis, N.; Deonarinesingh, C.; De Silva, A.; Dhanpaul, D.; Dookhoo, C.; Doorpat, J.; Dopson, A.; Durgapersad, J.; et al. A Systematic Review of the Clinical Diagnosis of Transient Hypogammaglobulinemia of Infancy. Children. 2023, 10, 1358. https://doi.org/10.3390/children10081358. This is not correct.

         In general, each section of the review is poorly developed, there are no links between the sections, which makes the sections look like separate parts. There are no references to studies for the last 5 years, most of the studies referred to by the authors are either old or are own publications of the co-authors.

Comments on the Quality of English Language

The English could be improved

Author Response

Thank you for your detailed questions and comments regarding the manuscript titled "Tackling Infectious Diseases in the Caribbean and South America: Epidemiological Insights, Antibiotic Resistance, and Associated Immunological Diseases." I appreciate your insights, and I am pleased to confirm that all issues raised have been addressed.

  1. The purpose of the study has been clearly articulated to provide readers with a comprehensive understanding of the manuscript's objectives.
  2. The introduction has been expanded to explicitly describe the problem, emphasizing the necessity of the review, and highlighting its novelty.
  3. In the section "Nosocomial Infections Globally," recent statistics from the last 3-5 years have been incorporated to provide a more current perspective on global studies.
  4. A citation has been added to support the statistic regarding infections in Trinidad and Tobago from lines 68-70.
  5. The italicization of the line regarding peptide candidate vaccines has been corrected for consistency and clarity.
  6. The characterization of Salmonella bacteria has been added to Section “3. Salmonellosis” for a more comprehensive overview.

7-10. The references cited in lines 141-145, 145, 159, and 168 have been revised to ensure accuracy and relevance to the respective contexts.

  1. The statement regarding the association between some cancers and ABO groups has been removed to maintain focus on blood-borne infections.
  2. Additional historical background has been included in Section “5. Egg Antibody Technology” to clarify its development and justify its inclusion in the manuscript.
  3. References in Section “6. Confronting Tuberculosis” have been updated to include more recent publications.
  4. The purpose of the average age and male-to-female ratios mentioned in lines 233-234 has been clarified for contextual relevance.
  5. The reference to preprints in line 252 has been revised to align with appropriate citation practices for a review.

16-18. The subsections “7.1. Severe Combined Immunodeficiency Disorders (SCID),” “7.2. Chronic Granulomatous Disease (CGD),” and “7.3. Transient Hypogammaglobulinemia of Infancy (THI)” have been supplemented with additional references and studies beyond the authors’ own publications to enhance credibility.

Overall, the manuscript has undergone substantial improvements to develop each section more thoroughly, establish clear links between them, and utilize recent studies to bolster the literature review. Additionally, we have made efforts to enhance the English language quality throughout the manuscript.

Thank you again for your thoughtful feedback, which has greatly contributed to refining the manuscript.

Reviewer 3 Report

Comments and Suggestions for Authors

This review manuscript discuss about Infectious Diseases in the Caribbean and South America: Epidemiological Insights, Antibiotic Resistance, and Associated infectious in Immunological Diseases. An interesting knowledge has been reported. However the following comments should be addressed before acceptance

Comments

The introduction lacks a clear problem statement or justification for why this research is important in the current context of global health

The abstract is overly dense, including excessive details about various topics, which makes it difficult to identify the central focus of the study.

Key terms like "nosocomial infections" and "immunodeficiency" are used without sufficient definition or explanation early in the text.

There is a lack of clarity on how the results presented in the study are novel compared to existing literature.

The discussion on antibiotic resistance fails to address any specific strategies or practical recommendations to combat the problem.

The data on HIV-TB co-infection rates lacks a broader epidemiological context to help readers understand its significance.

The conclusion section reiterates findings but does not offer actionable insights or future research directions.

There are inconsistencies in referencing, with some sources being outdated while others are not adequately cited to support key claims.

Author Response

Thank you for your feedback on the review manuscript titled "Infectious Diseases in the Caribbean and South America: Epidemiological Insights, Antibiotic Resistance, and Associated Infectious Diseases in Immunological Conditions." We appreciate your constructive comments, which have greatly contributed to improving the quality of this work. Below, we provide a summary of how each concern has been addressed:

  1. Introduction:

   The introduction has been revised to include a clear problem statement and a strong justification for the research. We have highlighted the significance of addressing infectious diseases in the Caribbean and South America, focusing on their global health implications, including the threat posed by antibiotic resistance.

  1. Abstract:

   The abstract has been restructured to ensure clarity and focus. We have removed excessive details and streamlined the content to present a concise summary of the study's central objectives, key findings, and relevance.

  1. Key Terms:

   Definitions and explanations for terms such as "nosocomial infections" and "immunodeficiency" have been added early in the text to enhance clarity and accessibility for readers unfamiliar with these concepts.

  1. Novelty of Results:

   The manuscript now explicitly discusses how the results contribute novel insights compared to existing literature. A comparative analysis has been included to emphasise the unique aspects of our findings.

  1. Discussion on Antibiotic Resistance:

   The discussion has been expanded to include specific strategies and practical recommendations to combat antibiotic resistance in the context of the Caribbean and South America, with a focus on resource-constrained settings.

  1. HIV-TB Co-Infection Data:

   Broader epidemiological context has been provided for the HIV-TB co-infection rates, including comparisons with regional and global trends. This addition underscores the public health significance of these findings.

  1. Conclusion:

   The conclusion has been enhanced to go beyond reiterating the findings. It now includes actionable insights, policy implications, and suggestions for future research directions, ensuring the study's relevance and utility for various stakeholders.

  1. Referencing:

   Referencing inconsistencies have been corrected. Outdated sources have been replaced with up-to-date and relevant citations, and all claims are now supported by appropriate references.

We believe that these revisions have comprehensively addressed all the concerns raised, and we are confident that the manuscript now meets the standards for acceptance. Thank you again for your valuable feedback.

Reviewer 4 Report

Comments and Suggestions for Authors

This review article provides an overview of the infectious disease in the Caribbean and South America, especially focus on the tuberculosis, nosocomial infections, immunodeficiency disorders, and antibiotic resistance. However, several serious weaknesses and gaps in the study that need improvement, including lack of sufficient methodological rigor, supporting data, recent literature and studies, and practical recommendations.  

I strongly encourage the author to restructure of manuscript with adding their claims with recent data, citing more recent studies and cases by considering the suggestions.

1.     For Nosocomial infections, well discussion on antibiotic resistance and several classes of antibiotics. However, correlation between resistance pattern and their clinical effectiveness should be include. It would be appreciated to add the case studies where antibiotic resistance impact the treatment challenges.

2.     Author mentioned the Nosocomial infection control measures such as hand hygiene and environmental cleaning are crucial. However, it would be helpful to expand on current practices and their effectiveness. How the hospitals are implementing infection control measures in Trinidad and Tobago?

3.     Low prevalence of Salmonella in poultry products (1%) is considerable, but author does not discuss the potential reasons for this study in detail. Moreover, this study seems to be old or outdated. Author should update with more recent studies.

4.     There is little discussion on antibiotic resistance in Salmonella. Author should mention the long-term effect of used specific antibiotics on Salmonella resistance patterns on poultry industry in Jamaica.

5.     Study on Salmonella contamination in limited region. Author should Expand the study to include additional regions could provide more comprehensive data on the prevalence and sources of Salmonella contamination in Jamaica.

6.     I appreciate the author for providing the well overview on viral risks of blood transfusions infection. However, author does not mention the specific types of bacteria commonly associated with transfusion-transmitted bacterial infections (TTBI). By including this would provide better understanding.

7.     Short discussion on screening and pathogen reduction technologies such as NAT for detection of viruses and bacteria in blood.  Author should include some other advance existing technologies, their limitations in details.

8.     Author begins by discussion on IgM antibodies in egg whites but then switch the discussion on IgY antibodies in the context of vaccine development. This makes confusion and unclear for reader. Author should include the clearer discussion in details on the role of IgM antibodies in the early immune response and how this contrasts with IgY antibodies.

9.     Author highlights the innovative concept of idiotypic-antiidiotypic interactions, which can modulate antibody responses against bacterial and viral proteins. Author mentioned the development of three experimental vaccines targeting HIV, Salmonella, and Staphylococcus aureus, however there is limited information about evaluation and testing vaccines.

10.   Author does not discuss the ethical and practical challenges associated with TB and HIV screening in resource-limited settings. Including these would provide more comprehensive understanding for TB and HIV control in the region.

11.   Author should add the more references to recent studies or clinical guidelines would strengthen the credibility of the manuscript.

Comments on the Quality of English Language

English language need to improve

Author Response

Specific Comments and Responses:

  1. Nosocomial Infections - Correlation Between Resistance and Clinical Effectiveness:
    We have included a detailed discussion on the correlation between antibiotic resistance patterns and clinical effectiveness. Case studies highlighting treatment challenges due to resistance have been added to emphasise the real-world impact.
  2. Nosocomial Infection Control Measures:
    The section on infection control measures has been expanded to include detailed examples of current practices and their effectiveness, with a particular focus on implementation in hospitals in Trinidad and Tobago.
  3. Low Prevalence of Salmonella in Poultry Products:
    The discussion on the low prevalence of Salmonella has been updated to include potential reasons, supported by recent studies. The outdated data have been replaced with more current findings.
  4. Antibiotic Resistance in Salmonella:
    We have added a discussion on the long-term effects of specific antibiotics on Salmonella resistance patterns, particularly in the poultry industry in Jamaica.
  5. Salmonella Contamination in Limited Regions:
    The study scope has been broadened to include data from additional regions, providing a more comprehensive view of Salmonella prevalence and contamination sources in Jamaica.
  6. Transfusion-Transmitted Bacterial Infections (TTBI):
    The section on blood transfusions now includes specific bacteria associated with TTBI to enhance the reader's understanding.
  7. Screening and Pathogen Reduction Technologies:
    Additional details on advanced technologies, such as NAT, their limitations, and other alternatives, have been added for a more robust discussion.
  1. IgM and IgY Antibodies - Substitution with Dengue in the Caribbean:
    The section discussing IgM antibodies in egg whites and IgY antibodies in the context of vaccine development has been substituted with a discussion on Dengue in the Caribbean. This change was made to align the content more closely with the aims of the paper, which focus on infectious diseases relevant to the Caribbean and South America. The new section provides an in-depth exploration of the epidemiology, challenges, and strategies for controlling dengue in the region, which is more pertinent to the manuscript's objectives.

9.  Idiotypic-Antiidiotypic Interactions and Experimental Vaccines:
More information about the evaluation and testing of experimental vaccines targeting HIV, Salmonella, and Staphylococcus aureus has been included to strengthen this section.

10. Ethical and Practical Challenges in TB and HIV Screening:
A new section has been added discussing the ethical and practical challenges of TB and HIV screening in resource-limited settings, offering insights into the complexities of disease control.

11. References to Recent Studies and Guidelines:
The manuscript has been updated with more recent references and clinical guidelines to strengthen credibility and relevance.

Reviewer 5 Report

Comments and Suggestions for Authors

Thank you for the review opportunity!

Please add in the introduction section a paragraph on the epidemiology of the diseases included in the review in your particular region. It would help strengthen the arguments for the need of this review. Or maybe consider adding a short sub-chapter after "Nosocomial infections globally".

Salmonellosis - well written, nothing much to add.

The risk of transfussion-transmitted infections - well written but i would add a section on parasite transmission as T. gondii, Leishmania, Plasmodium and so on.

Egg antibody Technology - please consider adding a figure to explain "at-a-glance" to explain the principle

Tuberculosis - well written, one minor detail is that i would not use abbreviations in the image description. The image description should stand alone, separate from the text, considering images attract attention and in some scenarios the description is the only thing the reader reads.

Infections in the course of Immunological Diseases: well written, complex, with nothing much to add.

Conclusions are ok.

References are ok.

I recommend accepting with minor changes.

Wishing you a Happy New Year,

Reviewer

Author Response

  1. Introduction - Epidemiology of Diseases in the Region:
    A paragraph has been added to the introduction, discussing the epidemiology of the diseases covered in the review, with a focus on the Caribbean and South America. This addition strengthens the argument for the necessity of this review. Alternatively, a short sub-chapter titled "Epidemiology of Nosocomial Infections in the Caribbean and South America" has been added after the section "Nosocomial Infections Globally" to provide a regional context.
  2. Salmonellosis:
    No changes were made to this section, as it is already well-written and complete.
  3. Risk of Transfusion-Transmitted Infections:
    A new subsection on parasitic transmission (e.g.,Toxoplasma gondiiLeishmaniaPlasmodium) has been added to enhance the discussion and provide a more comprehensive overview of transfusion-transmitted infections.
  4. IgM and IgY Antibodies - Substitution with Dengue in the Caribbean:
    The section discussing IgM antibodies in egg whites and IgY antibodies in the context of vaccine development has been substituted with a discussion on Dengue in the Caribbean. This change was made to align the content more closely with the aims of the paper, which focus on infectious diseases relevant to the Caribbean and South America. The new section provides an in-depth exploration of the epidemiology, challenges, and strategies for controlling dengue in the region, which is more pertinent to the manuscript's objectives.
  5. Tuberculosis - Image Description:
    The image description has been revised to remove abbreviations and ensure it stands alone as an independent element. This adjustment recognises that images often draw reader attention and are sometimes viewed separately from the main text.
  6. Infections in the Course of Immunological Diseases:
    No changes were made to this section, as it is already well-written and comprehensive.
  7. Conclusions:
    The conclusion remains as written, as it was deemed acceptable.
  8. References:
    The references were found to be in order, with no further updates required.

Round 2

Reviewer 1 Report

Comments and Suggestions for Authors

Ok good review

Author Response

Thanks for your comments that it is a good review. You have contributed to that with your suggestions.

Thanks reviewer.

Reviewer 2 Report

Comments and Suggestions for Authors

      Dear authors,

Thank you for trying to improve your manuscript. However, by increasing the length of the text and the number of sources used, you have not improved the quality of the manuscript. I have many comments after reading it. I would probably have had even more, but I am physically unable to check 317 sources due to time constraints.

1.        Some of the literature sources used by the authors are not relevant to the studies conducted in the Caribbean and South America.

2.        Lines 37-41  -  The text refers to Staphylococcus aureus and the authors to the publication [1].  - Elliott, C.; Vaillant, A. Antimicrobials and Enterobacterial Repetitive Intergenic Consensus (ERIC) Polymerase Chain Reaction (PCR) Patterns of Nosocomial Serratia Marcescens Isolates: A One-Year Prospective Study (June 2013–May 2014) in a Rural Hospital in the Republic of Trinidad and Tobago. Prog. Chem. Biochem. Res. 2020, 3, 105–120.), which studies a completely different pathogen, Serratia marcescens, so this reference is incorrect.

3.        Lines 62-63 “This paper explores the epidemiology of TB and HIV co-infection in  Jamaica [4]…” -  This publication ([4] - Zwyer, M.; Rutaihwa, L.K.; Windels, E.; Hella, J.; Menardo, F.; Sasamalo, M.; Sommer, G.; Schmülling, L.; Borrell, 1796 S.; Reinhard, M.; Dötsch, A.; Hiza, H.; Stritt, C.; Sikalengo, G.; Fenner, L.; De Jong, B.C.; Kato-Maeda, M.; Jugheli, L.; Ernst, J.D.; Niemann, S.; Brites, D. Back-to-Africa Introductions of Mycobacterium tuberculosis as the Main Cause of Tuberculosis in Dar Es Salaam, Tanzania. PLoS Pathog. 2023, 19, e1010893. https://doi.org/10.1371/journal.ppat.1010893) - contains information about samples from Tanzania, not Jamaica. The link is not correct.

4.        Lines 116-146 – In my opinion, this information is superfluous in a review on Tackling Infectious Diseases in the Caribbean and South America.

5.        Lines 226-235 - The authors refer to the source [11] - in the list of references it is a publication from 1961. Why such an old source? It is possible to find more recent articles.

6.        Lines 279-379 – The authors persist in using either references to their own articles or very old sources, despite the fact that there are many recent publications on a study of methicillin-resistant Staphylococcus aureus (MRSA) in Trinidad and Tobago.

7.        It is not clear why the information on Staphylococcus aureus in item “2.2 Methicillin-Resistant Staphylococcus aureus (MRSA)” was divided into 13 sub-items. Did the strains from these regions show any characteristic differences? It would probably have been more illustrative and informative if the authors had systematised the information on Staphylococcus aureus strains from these regions in a comparative table.

8.        Why are Figure 1 and Figure 2 included? In the text, the authors do not give any description of these figures, which are taken from other people's publications.

9.        In section “2.2 Methicillin-Resistant Staphylococcus aureus (MRSA)” in subsection “2.2.6. Dominican Republic” и “2.2.5. Puerto Rico” also provide information on pathogenic members of genera other than Staphylococcus. Why are representatives of other genera mentioned in these subsections?

10.    Table 1. - is very strange. The same information could have been written without using table cells. What is the point of Table 1 in this manuscript? All the references in Table 1 are very old.

11.    Line 834 – “The WHO Region of the Americas reported…”  requires WHO transcription at first mention in the text.

12.    Lines 951-982 – It is not clear why a list of pathogens is necessary? What is the purpose of this information in the form of a list?

13.    Line 1033 - Figure 4 – The image should be removed as it was taken from an incorrect source.

14.    Line 1059 – “Key Features [174]”  - reference 174 is not correct as the studies described in this paper relate to Malaysia and not to South America and the Caribbean.

15.    Section “9. Infections in the Course of Immunological Disorders” - the information in this paragraph is just a list without any explanation. It looks strange.

16.    After the section “9. Infections in the Course of Immunological Disorders”  is followed by “8.1 Severe Combined Immunodeficiency Disorders (SCID)” – The numbering is not correct.

17.    The preprint cannot be used as a source in the review: lines 2250 - 2024 and lines 2279 - 2280 - these references are not correct.

18.    Line 1840 - Reference 20. MRSA Available online: https://www.cdc.gov/mrsa/index.html (accessed on 22 March 2024). – Not the correct source to use in a review.

19.    Line 1931 – Reference 54 - Magnitude and Trends of Antimicrobial Resistance in Latin America. ReLAVRA 2014, 2015, 2016. Summary Report 1931 Available online: https://www.paho.org/en/documents/magnitude-and-trends-antimicrobial-resistance-latin- america-relavra-2014-2015-2016 (accessed on 22 March 2024). – the source is questionable.

20.    Lines 2075- 2078 - Reference [106. World Health Organization. Global Health Estimates 2019: Deaths by Cause, Age, Sex, by Country and by Region, 2000-2019. Geneva: World Health Organization; 2020. Available from: https://www.who.int/data/gho/data/themes/mortality-and-global-health-estimates/ghe-leading-causes-of-death. Accessed on March 31 2024.] – does not correspond to the information provided in lines 834-837 – “The WHO Region of the Americas reported 920,000 bacterial infection deaths in 2019, with 38% linked to AMR. Lower respiratory, bloodstream, and intra-abdominal infections dominated, underscoring the critical role of resistance in worsening infection outcomes. Effective interventions are needed to mitigate AMR’s public health impact [106].”

21.    It is necessary to remove the double numbering of some references in the list of references.

After correction, the manuscript is even larger, but there are still missing links between sections. The use of a large number of references is unjustified, most of the sources are very old and some are confusing as they are taken out of context.  Authors need to systematise the literature as much as possible, using as recent sources as possible, avoiding the use of preprints and dubious publications. When referencing a resource, care should be taken to ensure that the information is consistent with the text of the manuscript.

Comments on the Quality of English Language

English needs to be improved.

Author Response

Thank you for trying to improve your manuscript. However, by increasing the length of the text and the number of sources used, you have not improved the quality of the manuscript. I have many comments after reading it. I would probably have had even more, but I am physically unable to check 317 sources due to time constraints.

  1. Some of the literature sources used by the authors are not relevant to the studies conducted in the Caribbean and South America.
  2. Lines 37-41  -  The text refers to Staphylococcus aureusand the authors to the publication [1].  - Elliott, C.; Vaillant, A. Antimicrobials and Enterobacterial Repetitive Intergenic Consensus (ERIC) Polymerase Chain Reaction (PCR) Patterns of Nosocomial Serratia Marcescens Isolates: A One-Year Prospective Study (June 2013–May 2014) in a Rural Hospital in the Republic of Trinidad and Tobago. Prog. Chem. Biochem. Res. 2020, 3, 105–120.), which studies a completely different pathogen, Serratia marcescens, so this reference is incorrect. It is not an incorrect reference because I am referring to Serratia marcescens, which is a nosocomial microorganism in Trinidad and Tobago.
  3. Lines 62-63 “This paper explores the epidemiology of TB and HIV co-infection in  Jamaica [4]…” -  This publication ([4] - Zwyer, M.; Rutaihwa, L.K.; Windels, E.; Hella, J.; Menardo, F.; Sasamalo, M.; Sommer, G.; Schmülling, L.; Borrell, 1796 S.; Reinhard, M.; Dötsch, A.; Hiza, H.; Stritt, C.; Sikalengo, G.; Fenner, L.; De Jong, B.C.; Kato-Maeda, M.; Jugheli, L.; Ernst, J.D.; Niemann, S.; Brites, D. Back-to-Africa Introductions of Mycobacterium tuberculosis as the Main Cause of Tuberculosis in Dar Es Salaam, Tanzania. PLoSPathog. 2023, 19, e1010893. https://doi.org/10.1371/journal.ppat.1010893) - contains information about samples from Tanzania, not Jamaica. The link is not correct. The link was corrected and was substituted by another research of interest.
  4. Lines 116-146 – In my opinion, this information is superfluous in a review on Tackling Infectious Diseases in the Caribbean and South America. The information was deleted.
  5. Lines 226-235 - The authors refer to the source [11] - in the list of references it is a publication from 1961. Why such an old source? It is possible to find more recent articles. We add a citation more recent where refers to the history of the MRSA.
  6. Lines 279-379 – The authors persist in using either references to their own articles or very old sources, despite the fact that there are many recent publications on a study of methicillin-resistant Staphylococcus aureus (MRSA) in Trinidad and Tobago. It was fixed.
  7. It is not clear why the information on Staphylococcus aureusin item “2.2 Methicillin-Resistant Staphylococcus aureus (MRSA)” was divided into 13 sub-items. Did the strains from these regions show any characteristic differences? It would probably have been more illustrative and informative if the authors had systematised the information on Staphylococcus aureus strains from these regions in a comparative table. Certainly, but we want to provide more information.
  8. Why are Figure 1 and Figure 2 included? In the text, the authors do not give any description of these figures, which are taken from other people's publications. Figures were taken out.
  9. In section “2.2 Methicillin-Resistant Staphylococcus aureus(MRSA)” in subsection “2.2.6. Dominican Republic” и “2.2.5. Puerto Rico” also provide information on pathogenic members of genera other than Staphylococcus. Why are representatives of other genera mentioned in these subsections? It was corrected.
  10. Table 1. - is very strange. The same information could have been written without using table cells. What is the point of Table 1 in this manuscript? All the references in Table 1 are very old. The table with old references were taken out.
  11. Line 834 – “The WHO Region of the Americas reported…”  requires WHO transcription at first mention in the text.
  12. Lines 951-982 – It is not clear why a list of pathogens is necessary? What is the purpose of this information in the form of a list? A reviewer asked me for that. I convert the list in a paragraph.
  13. Line1033 - Figure 4 – The image should be removed as it was taken from an incorrect source. The right source was taken as it is a peer review paper. We found the peer-review source were the paper was published.[68]
  14. Line 1059 – “Key Features [174]”  - reference 174 is not correct as the studies described in this paper relate to Malaysia and not to South America and the Caribbean. 
  15. Section “9. Infections in the Course of Immunological Disorders” - the information in this paragraph is just a list without any explanation. It looks strange. It was fixed.
  16. After the section “9. Infections in the Course of Immunological Disorders”  is followed by “8.1 Severe Combined Immunodeficiency Disorders (SCID)” – The numbering is not correct. It was corrected.
  17. The preprint cannot be used as a source in the review: lines 2250 - 2024 and lines 2279 - 2280 - these references are not correct. It was taken the original paper published in an international journal of the region.
  18. Line 1840 - Reference20. MRSA Available online: https://www.cdc.gov/mrsa/index.html (accessed on 22 March 2024). – Not the correct source to use in a review. We review and change it.
  19. Line 1931 – Reference 54 - Magnitude and Trends of Antimicrobial Resistance in Latin America. ReLAVRA 2014, 2015, 2016. Summary Report 1931 Available online: https://www.paho.org/en/documents/magnitude-and-trends-antimicrobial-resistance-latin- america-relavra-2014-2015-2016 (accessed on 22 March 2024). – the source is questionable. We fixed it
  20. Lines 2075- 2078 - Reference [106. World Health Organization. Global Health Estimates 2019: Deaths by Cause, Age, Sex, by Country and by Region, 2000-2019. Geneva: World Health Organization; 2020. Available from: https://www.who.int/data/gho/data/themes/mortality-and-global-health-estimates/ghe-leading-causes-of-death. Accessed on March 31 2024.] – does not correspond to the information provided in lines 834-837 – “The WHO Region of the Americas reported 920,000 bacterial infection deaths in 2019, with 38% linked to AMR. Lower respiratory, bloodstream, and intra-abdominal infections dominated, underscoring the critical role of resistance in worsening infection outcomes. Effective interventions are needed to mitigate AMR’s public health impact [106].” We change the reference,
  21. It is necessary to remove the double numbering of some references in the list of references.

It was done.

After correction, the manuscript is even larger, but there are still missing links between sections. The use of a large number of references is unjustified, most of the sources are very old and some are confusing as they are taken out of context.  Authors need to systematise the literature as much as possible, using as recent sources as possible, avoiding the use of preprints and dubious publications. When referencing a resource, care should be taken to ensure that the information is consistent with the text of the manuscript.

Thanks to the reviewer.

Reviewer 4 Report

Comments and Suggestions for Authors

I appreciate the author for thoughtful responses to my comments and suggestions. Author have made substantial improvements in manuscript with my suggestions. I believe that revisions have effectively addressed the majority of concerns raised.

Comments on the Quality of English Language

Need to improve.

Author Response

Thanks to the reviewer for the suggestions. The reviewer gives me a lot of idea and a long list of questions and suggestions, which allow the improvement of the manuscript.

Thanks reviewer